behaviour, ecology, evolution

immigration, cultural adaptation, ecological conditions, evolution

**Author for correspondence:**
R. Lynch
e-mail: robertflynch@gmail.com

# Socio-cultural similarity with host population rather than ecological similarity predicts success and failure of human migrations

R. Lynch[1], J. Loehr[2], V. Lummaa[3], T. Honkola[3], J. Pettay[3] and O. Vesakoski[3]

[1]Pennsylvania State University, Department of Anthropology, 410 Carpenter Building, University Park, PA 16802, USA
[2]University of Helsinki, Biological and Environmental Sciences, Viikinkaari 1 PO Box 65, Helsinki, Finland
[3]University of Turku, Department of Biology, Vesilinnantie, 5, Turku 20014, Finland

(iD) RL, 0000-0002-2477-6204

Demographers argue that human migration patterns are shaped by people moving to better environments. More recently, however, evolutionary theorists have argued that people move to similar environments to which they are culturally adapted. While previous studies analysing which factors affect migration patterns have focused almost exclusively on successful migrations, here we take advantage of a natural experiment during World War II in which an entire population was forcibly displaced but were then allowed to return home to compare successful with unsuccessful migrations. We test two competing hypotheses: (1) individuals who relocate to environments that are superior to their place of origin will be more likely to remain—*The Better Environment Hypothesis* or (2) individuals who relocate to environments that are similar to their place of origin will be more likely to remain—*The Similar Environment Hypothesis*. Using detailed records recording the social, cultural, linguistic and ecological conditions of the origin and destination locations, we find that cultural similarity (e.g. linguistic similarity and marrying within one's own minority ethnic group)—rather than ecological differences—are the best predictors of successful migrations. These results suggest that social relationships, empowered by cultural similarity with the host population, play a critical role in successful migrations and provide limited support for the similar environment hypothesis. Overall, these results demonstrate the importance of comparing unsuccessful with successful migrations in efforts understand the engines of human dispersal and suggest that the primary obstacles to human migrations and successful range expansion are sociocultural rather than ecological.

## 1. Introduction

> *Only a fool learns from his own mistakes. The wise man learns from the mistakes of others*
> *-Otto von Bismarck* [1]

Research seeking to understand how the environment affects human migration has been focused almost exclusively on successful migrations. In human genetics, for example, an ever increasing number of studies have used the distribution of both modern and ancient DNA samples [2–4] to infer migration routes of early humans. At the same time, cultural studies have focused on understanding dispersal routes using archaeological sites [5], cultural phylogenies [6] and language data—e.g. the Bantu expansion [7,8], dispersion of South- and North-American languages [9] and Indo-European languages [10,11]. Despite tremendous progress investigating these questions, most of these studies rely on successful outcomes because analyses are based exclusively

on the contemporary genetic, cultural and or linguistic variation of populations that ultimately populated these areas—unsuccessful attempts (see electronic supplementary material: Failed prehistoric migration attempts) left no record. Therefore, there is a pressing need to compare unsuccessful migrations with those that succeeded. This will not only provide insight into how humans were able to successfully inhabit a diverse range of habitats and expand their range across most of the planet [12] but will also shed light on how dispersals occur and offer insight into contemporary immigration patterns today. In other words, a complete picture of what drives human range expansion requires an analysis of failed migration attempts and an understanding of the reasons for their failure. There is also a need for studies to distinguish between primary migrations (i.e. a founding population), and secondary migrations (i.e. those who came after this initial wave) where humans already live—each of which may depend on different skills and adaptations. Finally, much of this research has failed to systematically compare the conditions between the origin and destination (see [13]). The central purpose of this paper is to try to understand how both ecological and social factors in the origin and destination locations affect the success and failure of secondary migrations.

The environment has long been seen as an important driver of human dispersal [14] and there are two competing theories—demographic [15] and evolutionary [16,17]—on how local ecology influences immigration. Ravenstein [18] was the first demographer to attempt to discern the laws governing migration and he identified economic motives and the desire for humans to better themselves as the primary cause. Recently, research has modelled how deteriorating environments such as those brought on by climate change [19] and land degradation [20] affect migration and it is generally accepted that changing environments can impact dispersal patterns [13,21]. The importance of ecological conditions may also extend across species. Barsbai et al. [22], for example, showed that foraging, reproductive and social behaviours of humans, birds and mammals living in the same areas are remarkably similar. Lee [15] tried to develop a comprehensive framework around which factors predict migration direction and flow. He observed that migration tends to take place along well-defined routes with people from similar backgrounds forming outposts (i.e. primary migration), reporting back and recruiting their fellows (i.e. secondary migration) and noted that for every major migration stream, a strong counterstream (i.e. reverse migration) develops if migrants are pushed out of a place of origin by poor conditions rather than pulled into a new destination by better conditions. Lee hypothesized that the push–pull forces of both 'stream and counterstream tend to be low if origin and destination are (ecologically and culturally) similar.' [15, p. 55]. In this interpretation, humans are expected to move to environments that are superior.

Evolutionary theorists meanwhile have proposed that human migratory success is likely linked to cultural adaptations [17] that allow us to acquire locally adaptive behaviour in a wide range of environments and are leveraged by our ability to learn from each other. For secondary migrations, this can mean adopting local knowledge which, when ignored, can have severe costs [23]. Theories based on cultural adaptations and evolution differ from demographic theories and are unique, however, in their prediction that there will be costs of moving to better environments if a population is culturally adapted to a particular way of life and is either unable or unwilling to learn and adapt to a new one [24]. Indeed, Bazzi et al. [25] showed that villages in Indonesia which were assigned a higher proportion of migrants who had experienced similar climate and soil conditions achieved greater rice production, indicating that similar agro-climatic conditions are important for their transferability. Consistent with predictions of cultural evolution, they also showed that interactions with the host population and social capital in the resettlement areas were important predictors of rice productivity.

Although cultural practices are generally learned and are frequently tied to ecological conditions (e.g. cross-cousin marriage practices may depend on a population's subsistence strategy) [26], the forms of words themselves have no fitness implications so linguistic differences are often used as neutral markers to track the diverging cultural histories of populations [27]. Using linguistic differences between municipalities in Finland, Honkola et al. [28] found that diverging dialects were better predicted by ecological and cultural differences than geographical distance. This supports the view of cultural adaptation proponents who argue that cultural practices respond to challenges posed by particular environments [29]. The fact that these differences in dialects, which are likely to reflect cultural differences, have been maintained despite constant human movements between these regions, suggests that there may be factors enforcing the cultural segregation of these populations. Other sociocultural and economic factors like tax rates, however, are more likely to elicit directional preferences with individuals preferring lower taxes [30]. However, because tax rates are strongly tied to income, we might also expect individuals to prefer areas with higher taxes and therefore higher productivity. Nevertheless, if cultural practices are tied to ecological conditions as expected, then cultural pre-adaptations—pre-existing cultural knowledge—will help to ensure successful immigration to areas with similar ecological conditions. Indeed, there is evidence that new immigrants cluster in areas that are culturally similar [31,32], which can exacerbate the correlation between sociocultural and ecological similarity. The importance of social networks in immigration [33,34] and spatial assortment by ethnicity and nationality, known as 'immigrant enclaves', highlights how critical the sociocultural similarity of the host population is for decisions on where to migrate. People tend to move to places where people like them have already settled [35,36] which suggests that existing social networks are important for the success of new migrants. If this line of reasoning is correct, then humans might be expected to successfully move to environments that are more similar to those to which they are accustomed or culturally adapted.

Understanding the relative importance of ecological conditions and one's ability to culturally adapt to a new place is difficult because many attempted migrations are likely to have ended in failure, and recent advances in ancient DNA sequencing have shown a number of early migrations into Europe and central Asia that left no modern day descendants [37]. This not only leaves an important part of the human migration story untold but also leaves the hypothesized engines of human dispersal vulnerable to heavily filtered data. A comparison of successful with unsuccessful migrations, however, would shed light on the patterns we currently see in languages, genes and the archaeological record and would enable researchers to better understand the general processes that affect human dispersal overall.

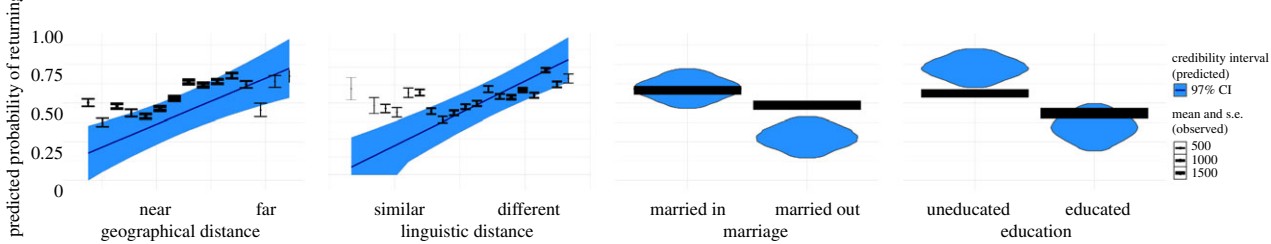

**Figure 1.** Similar environment hypothesis for non-farmers. Closer geographical distance between the origin and destination locations, greater linguistic similarity, marrying someone from the host location (i.e. western Finland) and having a job that requires an education all strongly predict the likelihood of a non-farmer evacuee remaining in the host location. Model generated posterior distribution predictions (dark lines); credibility intervals (blue shading) drawn from the top model in model comparisons (see electronic supplementary material, figure S5 for posterior distributions for all covariates in the model). The observed data (means and standard errors) are also shown with samples less than 50 removed. (Online version in colour.)

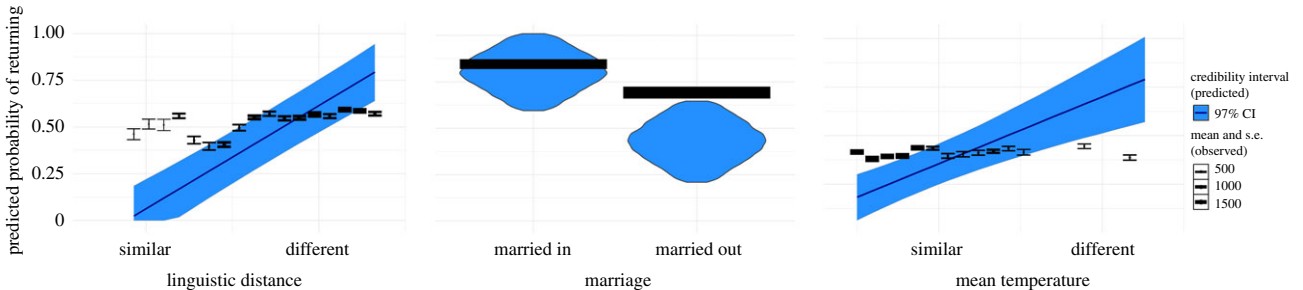

**Figure 2.** Similar environment hypothesis for farmers. Greater linguistic similarity, marrying someone from the host location (i.e. western Finland) and similar temperatures between the origin and destination locations all strongly predict the likelihood of a farmer evacuee remaining in the host location. Model generated posterior distribution predictions (dark lines), credibility intervals (blue shading) drawn from the top model in model comparisons (see electronic supplementary material, figure S5 for posterior distributions for all covariates in the model). The observed data (means and s.e.) are also shown with samples less than 50 removed. (Online version in colour.)

Here, we use an unusually well-documented dataset recording the movements and occupations of a population of evacuees from Karelia (electronic supplementary materials, figure S1), Finland during World War II to analyse which factors best predict successful dispersal. A unique historical situation that occurred in Finland during World War II, in which an entire population was forced to evacuate but then allowed to return home (see electronic supplementary material, Methods: historical background) provides a quasi-natural experiment which allows us to analyse the characteristics of the individuals, the places they moved to and from, and whether they returned. Previous work, using a different dataset on this same population, showed that habits contributed to farmers remaining in agriculture after the war [38]. This analysis, however, relied on the strength of social networks within the evacuee population whereas we are primarily interested in relationships between evacuees and the host population (see electronic supplementary material, Discussion). To our knowledge, data on the same immigrant population has never been used to evaluate unsuccessful migrations, so these data present an extraordinary opportunity to investigate which factors affect the likelihood of success. By comparing extraordinarily rare and detailed data quantifying the ecological, linguistic, cultural and geographical conditions at the origin and destination locations of an entire population and linking them to the personal characteristics and movements of the evacuees themselves, we are able to analyse how different types of environments affect migration success.

We compare two subsets of data grouped by occupation—farmers and non-farmers—using non-farmers as a control group but focusing on the movements of farmers both because agricultural production is closely tied to ecological conditions and because farming techniques are culturally transmitted (see

Methods: data). In other words, the success of non-farmers depends more on sociocultural factors than ecological conditions such as soil conditions whereas the success of farmers depends on both ecological and socio-cultural factors. Using pre-registered hypotheses [39], we distinguish between demographic hypotheses that successful migrations are predicted by individuals moving to better environments, and evolutionary hypotheses that successful migrations will be predicted by individuals moving to similar environments to which they are culturally adapted. The particular ecological conditions that we considered to be 'better' were largely determined by the preferences of farmers when compared with non-farmers (see Methods), as optimal conditions (e.g. more or less rain) largely rely on the type of crop. Specifically, we predict that (1) individuals who move to environments that are superior to the environments in which they were living in Karelia (e.g. better soil or longer growing season) will be more likely to remain in their relocation municipality—*The Better Environment Hypothesis* or (2) that individuals who move to environments that are both culturally and ecologically more similar to their origin location in Karelia will be more likely to remain in their relocation municipality—*The Similar Environment Hypothesis*.

## 2. Results

Overall, we found almost no support for the *Better Environment Hypothesis* and some support for the *Similar Environment Hypothesis* (table 1). All models, however, revealed the importance of sociocultural similarity between regions in predicting migration success (figures 1 and 2). Across all models, being younger, male, marrying outside of one's ethnic group at any

time (i.e. either before or after the war), being educated, and evacuating to a linguistically more similar municipality are all strongly and positively associated with the likelihood of remaining in the evacuation destination. (For effects of age, sex and number of children see electronic supplementary material, Results: personal characteristics.)

The top ranked models testing the *Better Environment Hypothesis* and the *Similar Environment Hypothesis* (see electronic supplementary material, Methods: model selection) were compared against models using all available variables for the non-farmers and the farmers datasets, respectively. In both datasets what we would expect to observe if we took an infinite number of future observations from the same data generating mechanism as our model (i.e. the expected log pointwise predictive density (ELPD)) the likelihood was highest for models testing the *Similar Environment Hypothesis* [H2]. Results show that the models which entered predictors as absolute differences and were therefore designed to test [H2] decidedly outperform those which entered predictors as directional differences and were therefore designed to test [H1] (see electronic supplementary material, table S2). We include the results for the top ranked models (the similar environment hypothesis) for non-farmers and farmers, respectively, testing each hypothesis alongside those with all variables entered because both approaches have drawbacks. On the one hand entering all of our predictors can lead to overfitting [40] and model selection can avoid pitfalls associated with entering variables unrelated to the outcome [41], while on the other, using model selection criteria can bias the ecological validity of hypotheses that were both pre-registered [39] and chosen based on our reading of the literature on which soil conditions and ecological factors are likely to affect the livelihood of farmers.

## (a) Sociocultural factors

Sociocultural differences (i.e. variables presumed to be primarily driven by human interactions, see Methods) with the host population, especially linguistic differences and marrying within a minority ethnic group that is not part of the host community were the most important obstacles to successful migrations. This provides some support for [H2] that social networks and connections with other humans are likely important for successful migrations. We tested whether marrying someone born outside of one's birth region predicted migration success because marrying into the host society would likely expand one's social network to include the relatives and friends of one's spouse in the migration area. Marrying someone from the host society was positively associated with remaining: for non-farmers the probability of remaining increases by 7% (49 to 57%), while for farmers it increases by 11% (75 to 86%) if they marry someone from western Finland and this effect does not seem to depend on whether they were married before or after the evacuation (see interaction between marrying out and marrying before the evacuation in table 1).

Having an education (available for non-farmers only: see Methods) was also positively associated with remaining and led to a predicted 10% increase in the probability of remaining (from 44% to 54%). Finally, linguistic difference was a strong and consistent predictor across all models of remaining. Moving to a linguistically more similar municipality

(25th to 75th percentile) resulted in a 7% decrease (58% to 51%) and a 5% decrease (86% to 81%) in the probability of returning for non-farmers and farmers, respectively.

A slightly less consistent factor predicting the likelihood of remaining (e.g. it was not included in the top ranked model using farmers) was geographical distance, and evacuating further from one's home reduced the probability of remaining. For non-farmers, moving further away is expected to increase the probability of returning by 4% such that those who moved 376 km (75th percentile) had a 56% probability of returning while those who only moved 228 km (25th percentile) were predicted to have a 52% probability of returning. It is important to note, however, that all these sociocultural factors mentioned above reflect absolute distance because they cannot easily be characterized as being better or worse (e.g. ingroup marriages, dialect differences and education). Tax rate and population density, however, do reflect directional differences because they can be either lower or higher in each location.

Municipalities with lower taxes and lower *per capita* income (see table 1 and electronic supplementary material, table S4, respectively) both seem to be preferred by non-farmers and farmers alike [H1] so these preferences are unlikely to be specific to the needs of farmers. Finally, population density, which also might be predicted to impact the decisions of all evacuees, does not seem to influence returning for either farmers or non-farmers in any of the models.

## (b) Ecological conditions

Ecological conditions, such as the type of soil, either have no directional [H1] (i.e. *Better Environment Hypothesis*) or absolute [H2] (i.e. *Similar Environment Hypothesis*) effects on the migration decisions of evacuees to reverse migrate. Ecological conditions also had similar effects on both farmers and non-farmers which suggests that these relationships are spurious because most of the ecological factors are expected to be particular to the needs of farmers (e.g. why would soil type affect migration decisions of urban factory workers?). It is however important to recognize that these variables may have indirect effects on the entire region which may then act upon farmers and non-farmers similarly (see electronic supplementary material, Methods: validating predictors). Ecological factors were entered both as the raw difference between the origin and destination location for a particular characteristic for testing [H1] and as the absolute distance for testing [H2]. Although all directional preference estimates [H1] that were either positive or negative and did not overlap with zero were flagged, only positive intervals were flagged for [H2] (table 1). This is because negative coefficients in the models testing the similar environment hypothesis were neither predicted (see Pre-registration [39]) nor easily interpreted as there is no reason to expect a cultural adaptation for unfamiliar conditions (e.g. for different soil types or temperatures). In other words, although it is reasonable to expect that farmers might prefer soil with either more or less peat or a soil type similar to what they were familiar with in Karelia, it is unclear why they would exhibit a preference for a soil composition that was simply different.

All of the ecological predictors were expected to be more relevant to the livelihoods of farmers. However, in models designed to test the better environment hypothesis [H1], none of them seem to have had much of an effect on the likelihood of farmers remaining. Instead, temperature, altitude,

**Table 1.** Parameter estimates, highest density intervals (HDI's) for factors affecting the likelihood of returning to Karelia for non-farmers (top panel) and farmers (bottom panel) for *the better environment hypothesis* (left side) and *the similar environment hypothesis* (middle) and top model (right). Geographical distance and sociocultural factors (i.e. linguistic similarity, marrying outside of ones group both before and after the war and education for non-farmers) are the best predictors of remaining in the evacuation destination while ecological factors such as soil types and rainfall do not consistently predict the likelihood of reverse migration. Parameter estimates are in italics if 95% HDI does not overlap with zero and are in the predicted direction (i.e. positive estimates for models testing *the similar environment hypothesis*).

| dependent variable | predictor | factor type | better environment | | similar environment | | top model: (similar environment) | |
|---|---|---|---|---|---|---|---|---|
| | | | 2.5% HDI | 97.5% HDI | 2.5% HDI | 97.5% HDI | 2.5% HDI | 97.5% HDI |
| returned to Karelia | intercept | | −1.02 | 0.21 | *0.18* | *0.57* | −0.03 | 0.39 |
| | age | personal characteristics | 0.99 | 1.47 | 0.96 | 1.45 | 0.98 | 1.39 |
| | sex | personal characteristics | −0.43 | −0.27 | −0.42 | −0.26 | −0.42 | −0.24 |
| | married before evacuation | personal characteristics | −0.09 | 0.17 | −0.10 | 0.16 | — | — |
| | geographical distance | geographical | 0.01 | 0.66 | 0.32 | 1.25 | 0.42 | 1.05 |
| | linguistic distance | sociocultural | 0.70 | 1.32 | 0.32 | 1.00 | 0.50 | 0.99 |
| | married out | sociocultural | −0.40 | −0.22 | −0.39 | −0.21 | −0.40 | −0.22 |
| | married out X married before evacuation | sociocultural | −0.32 | 0.14 | −0.31 | 0.16 | — | — |
| | education[a] | sociocultural | −0.52 | −0.29 | −0.50 | −0.26 | −0.49 | −0.27 |
| NON-FARMERS | taxes[b] | sociocultural | −2.06 | −1.36 | −1.16 | −0.80 | −1.19 | −0.86 |
| | population density | sociocultural | −0.26 | 0.74 | −0.31 | 0.06 | −0.18 | 0.52 |
| [N = 12 204] | | | | | | | | |
| *specific to the livelihood of farmers* | mean temperature | ecological | *−1.60* | *−0.53* | −0.80 | 0.12 | *−0.52* | *0.03* |
| | median altitude | ecological | *−1.65* | *−0.54* | −0.52 | 0.43 | — | — |
| | lake % | ecological | *0.22* | *0.85* | −0.34 | 0.08 | — | — |
| | livestock no. | ecological | *0.09* | *0.83* | *0.12* | *0.75* | *0.09* | *0.56* |
| | farmed % | ecological | *0.56* | *1.72* | −0.35 | 0.12 | — | — |
| | rainfall | ecological | −0.25 | 0.27 | −1.57 | −0.89 | −1.50 | −0.96 |
| | clay % | ecological [soil type] | −0.47 | 0.20 | −0.42 | 0.04 | — | — |
| | rock % | ecological [soil type] | −0.07 | 0.53 | −0.21 | 0.27 | — | — |
| | moraine % | ecological [soil type] | *0.83* | *1.56* | −1.04 | −0.54 | −1.01 | −0.60 |
| | peat % | ecological [soil type] | *0.00* | *1.02* | *0.19* | *0.84* | *0.29* | *0.80* |
| dependent variable | predictor | factor type | better environment | | similar environment | | top model: (similar environment) | |
| | | | 2.5% HDI | 97.5% HDI | 2.5% HDI | 97.5% HDI | 2.5% HDI | 97.5% HDI |
| returned to Karelia | intercept | | −0.98 | 1.21 | *0.24* | *1.19* | *0.24* | *1.19* |
| | age | personal characteristics | 1.61 | 2.31 | 1.61 | 2.19 | 1.62 | 2.20 |
| | sex | personal characteristics | −0.26 | −0.01 | −0.26 | −0.01 | −0.24 | −0.02 |
| | married before evacuation | personal characteristics | 0.10 | 0.42 | 0.07 | 0.40 | 0.10 | 0.36 |
| | geographical distance | geographical | 0.31 | 1.48 | 0.52 | 2.09 | — | — |
| | linguistic distance | sociocultural | 0.30 | 1.35 | 0.14 | 1.29 | 1.11 | 1.65 |
| | married out | sociocultural | −0.79 | −0.56 | −0.81 | −0.53 | −0.79 | −0.56 |

*(Continued.)*

**Table 1.** (Continued.)

| | | | | | | | | |
|---|---|---|---|---|---|---|---|---|
| | married out X | | | | | | | |
| | married before evacuation | sociocultural | −0.43 | 0.40 | −0.40 | 0.44 | — | — |
| | education[a] | sociocultural | — | — | — | — | — | — |
| FARMERS | taxes[b] | sociocultural | −2.49 | −1.10 | −1.65 | −0.84 | −1.60 | −0.95 |
| | population density | sociocultural | −0.78 | 1.08 | −0.78 | 0.99 | −0.69 | 0.75 |
| [N = 9870] | | | | | | | | |
| specific to the livelihood of farmers | mean temperature | ecological | −1.58 | 0.19 | 0.10 | 1.61 | 0.81 | 1.95 |
| | median altitude | ecological | −1.58 | 0.32 | −1.82 | 0.20 | −1.88 | −0.55 |
| | lake % | ecological | 0.60 | 1.40 | −0.06 | 0.64 | — | — |
| | livestock no. | ecological | −0.12 | 1.17 | −0.01 | 1.09 | −0.16 | 0.69 |
| | farmed % | ecological | −0.03 | 1.87 | −0.80 | 0.08 | — | — |
| | rainfall | ecological | −1.00 | −0.04 | −1.40 | −0.19 | −1.20 | −0.27 |
| | clay % | ecological [soil type] | −0.64 | 0.59 | −0.53 | 0.22 | — | — |
| | rock % | ecological [soil type] | −1.58 | 0.32 | −0.56 | 0.29 | −0.46 | 0.22 |
| | moraine % | ecological [soil type] | 0.64 | 1.88 | −0.51 | 0.23 | −0.57 | 0.13 |
| | peat % | ecological [soil type] | 0.17 | 1.88 | −0.78 | 0.17 | — | — |

[a]Education only entered for models using non-farmers.

[b]*Per capita* income and taxes were highly correlated and therefore could not be entered into the same models (see electronic supplementary material, table S4 for models using *per capita* income instead of taxes).

lakes, number of livestock, percentage of land farmed and both moraine and peat percentage in the soil all show directional effects [**H1**] for non-farmers (table 1) which suggests that these variables are not capturing the specific cultural adaptations and skills of farmers. Therefore, the only variable that offers any support for [**H1**] is annual rainfall as results indicate that farmers may have a slight preference for less rain overall. In the models testing the *Similar Environment Hypothesis* [**H2**], the only ecological factor that seemed to have influenced the likelihood of remaining was mean temperature. Results from both models, the model including all predictors and the top ranked model, both indicate that farmers prefer temperatures similar to those they experienced in Karelia. (See electronic supplementary material, table S5 for descriptive statistics (i.e. range and median values) for all continuous variables used in the study.)

## 3. Discussion

Overall results of models testing the effect of differences between the origin and the destination locations on migration success support the *Similar Environment Hypothesis* (**H2**)— successful migrations are to environments that are more similar—and provide little evidence to support *The Better Environment Hypothesis* (**H1**)—successful migrations are to environments that are superior. The most interesting finding, however, was that the best predictors of a successful migration (i.e. remaining in an evacuation location) across all models were sociocultural factors such as moving to a place with a more similar dialect and marrying outside of one's ethnic group. Meanwhile ecological factors, such as the amount of rainfall or the amount of peat seem to have little effect on decisions to remain. In general, these results suggest that sociocultural similarity is the key to successful

migrations and provide support for the *Similar Environment Hypothesis* [**H2**]. Theories grounded in cumulative cultural evolution predict that social learning is critical for human survival and that there are costs of moving to ecologically superior environments when populations are culturally adapted to specific ways of life and unable to adapt to new ones by learning from others [24]. Therefore, these results which show that sociocultural similarity between the origin and destination locations is important for successful migrations lends support to the cultural niche hypothesis. Overall, they suggest that cultural affinity with a host population can lead to more and or stronger social networks which then enable new migrants with the opportunity to learn from others and adapt to new environments.

Linguistic similarity with the host municipalities' dialect was a strong predictor of successful migrations for both farmers and non-farmers. Although language is seen as a neutral cultural trait because different dialects, words and accents are unlikely to have fitness implications in themselves [27], the ability to communicate and build ties with one's neighbours is likely to have important effects on fitness. In the previously mentioned study of Indonesian migrants, a one standard deviation increase in linguistic similarity between hosts and migrants predicted an increase in rice production by 25.8 per cent [25]. Experimental research and empirical findings indicate that accent and dialect are some of the strongest predictors of trust [42] and the likelihood of developing social bonds [43]. These findings support the hypothesis that developing social networks and enlisting the help of one's new neighbours are important for successful migrations. Linguistic differences between populations are thought to emerge when contacts between them become less frequent [44]. This is likely to occur when (1) populations differ ecologically and culturally [45], (2) cultural similarity determines migration routes such that individuals and groups that are

culturally adapted to particular environments seek out these places when they disperse [46,47] and/or (3) decisions to remain in a new location depend on how similar the new environments are to those from which they emigrated [15]. Results showing that successful migrations are likely to depend on the linguistic similarity of the origin and destination locations support these hypotheses while also suggesting that the ability to develop social ties and trust networks are also important for developing and maintaining linguistic differences between populations.

Marrying someone from the host population was another important sociocultural factor that had a strong and positive effect on remaining in the evacuation destination. Intermarriage between migrant and host populations is a commonly used metric to assess bridging social ties between migrants and hosts and overall social integration [48]. It is also commonly viewed by immigrants as the final step in the process of integration [49]. It is interesting to note that the positive relationship between marrying into the host population and remaining did not depend on when the marriage occurred and was true for individuals who married both before and after the evacuation. This suggests that the integrating effect of marrying into the host population is not simply the result of new ties to in-laws and the other social networks of one's native-born spouse, but rather may be a property of the individual. In other words, individuals who are more outwardly directed are simply more likely to form bridging relationships with the natives in general and this propensity may also cause them to marry into the host population.

Younger farmers and non-farmers of all ages were both more likely to successfully migrate. If older people have more fixed social networks and younger people are more flexible as previous research on this population suggests [50], then this also supports the hypothesis that the ability to develop ties to the host population are important for successful migrations. Younger people are more likely to have social networks that bridge group boundaries [51] and may therefore be more capable of picking up new skills and acquiring a whole body of novel cultural traits that make them competent in a new social or ecological environment.

Evacuees that were relocated further away from their homes were more likely to return. On the face of it, this is not what one might expect if geographical distance was the only factor influencing decisions to return (i.e. being displaced to more distant places should make returning more difficult). Some research, however, suggests that there is a positive relationship between geographical distance and cultural distance [52]. Like linguistic differences, cultural differences are likely to emerge when contacts between populations become less frequent and when it takes longer to travel somewhere these differences are more likely to take root. So even though we do not classify geographical distance as a sociocultural variable here, it may serve as a proxy for cultural distance which might help to explain why evacuees who relocated to more distant locations were more likely to return. This is because people arriving from more distant locations are likely to be more culturally different from the host population and may therefore have found it more difficult to assimilate and learn (e.g. acquire new farming techniques) from the host population.

The other sociocultural factors that both predicted a higher likelihood of remaining were having an education and either lower taxes or lower per capita income. Education

is commonly associated with increasing bridging social ties between different groups of people [53] and more educated migrants are likely better able to form attachments with the host community. Lee [15] also noted that more educated people in the professional and managerial classes tend to be positively selected to migrate. Taxes and per capita income are likely to be directional effects (i.e. Better Environment Hypothesis [H1]) and both seem to be preferred by farmers and non-farmers alike.

The only ecological characteristics that seemed to have any predictive value in our models were temperature and rainfall. Farmers seem to prefer mean temperatures that are more similar to those they experienced in their birth municipalities. Meanwhile rainfall, like taxes, is more likely to be a directional effect (i.e. Better Environment Hypothesis [H1]) such that farmers in Finland seem to prefer less rainfall overall. It might be argued, however, that because farming practices result from specific ecological contexts and that these practices are often learned from neighbours (i.e. via social learning), that it is difficult to parse these two influences and determine which is more important—sociocultural similarity or more similar ecological conditions. In this interpretation what is considered 'better' may be simply what the farmers are more used to. We are, however, able to distinguish between these possibilities in two ways. First, in the old environment the farmer learned how to grow specific crops in specific conditions in a relatively homogeneous population (i.e. the same ethnicity, dialect and culture) whereas in the new environment, he or she has to learn how to do this from his new neighbours who vary along these dimensions. We are therefore able to leverage this variance to determine the impact of socio-cultural similarity on the likelihood of remaining in the evacuation destination. Second, none of the ecological factors that are almost certainly indicative of worse soil conditions for all farmers, such as more rocks in the soil have any impact on decisions to remain. Although the top models support the similar environment hypothesis, overall, it is hard to argue that ecological conditions matter very much at all, regardless of whether they are classified as better or similar and suggests that regardless of the extent of the ecological differences, it is sociocultural familiarity that is critical for success.

These results may shed light on an important debate within evolutionary psychology over whether human success results from our unique cognitive abilities—the cognitive niche [54,55] or from gradually accumulated cultural information transmitted across generations—the cultural niche [24]. While the cognitive niche hypothesis proposes that human migratory success is driven by our singular ability to use cause-and-effect reasoning and to test and continuously refine behaviour within the lifetime of the individuals, the cultural niche hypothesis proposes that specific cultural information accrued and transmitted across many generations was needed. Although both hypotheses acknowledge the importance of the ability to learn from others, the cultural niche hypothesis relies more heavily on the importance of social learning.

Several important caveats are worth noting, however. First, our interpretation is only relevant for understanding secondary migrations where social contacts with the host population enable newcomers to the area to rapidly acquire the cultural adaptations required to flourish. For primary migrations and founder populations, this is not the case because there is no one to learn from. Secondly, ecological effects may be less important in east–west migrations like

those from Karelia to western Finland than they are along the north–south axis where temperature and sunlight differences are greater. Another limitation of the current study and the generalizability of these results to all migrants involves the specific circumstances in which Karelians were forcibly displaced by an invading army. While this allows us to more easily interpret these results (everyone was forced to leave regardless of any desire to migrate), the involuntary nature of the displacement is certain to have had an impact on the psychological motivations of the evacuees to return [56,57]. Future studies using qualitative data that could be systematically coded assessing reasons why individuals report returning or remaining would help to better understand and interpret these results. Qualitative data might also help add insight into how individuals who remained may have adapted farming practices to local conditions. Still, it is important to note that decisions to return or remain are likely to capture, in large part, a failure to adapt quickly to new conditions which in the harsh climate of Finland (e.g. short growing season and extreme temperatures) can result in the real possibility of total crop failure. It should be noted that because the ecological data are collected at the level of the municipality, the individual variation across plots of lands within these municipalities is obscured. Here, we are assuming that the ecological conditions of the individual plots are strongly associated with those of the surrounding areas. While this is almost certainly true of some conditions, including rainfall, temperature, and altitude, this may not always be the case with respect to the specific soil conditions. Finally, it should also be mentioned that another study that asked why so many farmers left agriculture in the decades after the war using a different dataset from the same population came to a different conclusion (see electronic supplementary material, Additional caveats).

## 4. Conclusion

Researchers have also long noted that social networks are important for the success of new migrants. But to our knowledge, no study has ever used a natural 'experiment' in which an entire population is forcibly displaced and then almost immediately after being evacuated is allowed to return home to test how cultural and ecological differences affect migration flow. Results suggest that the social networks and a similar culture predict successful evacuations while ecological conditions do not seem to matter as much for secondary migrations. Cultural adaptation is characterized by the accumulation of learned skills that help people to survive in particular environments, but which are not easily transferable to different environments. These findings provide evidence supporting this view and suggest that humans may be able to successfully adapt to new locations so long as they are able to form connections with and learn from the local population. We hope that these results will encourage researchers to consider how ecological differences between the origin and destination can be mitigated when migrants share a cultural background and have overlapping social networks with the host population. These findings are of general relevance for multidisciplinary efforts to understand how non-founder populations adapt to new environments and for understanding human migration patterns more generally.

## 5. Material and methods

These methods and statistical analyses were pre-registered and time stamped on the Open Science Framework on 8 February 2019 which was prior to our ability to access most of these data and prior to any analysis. The predictor variables, outcome variables and proposed analyses outlined below are nearly identical to those identified in the Open Science Framework pre-registration [39]. All discrepancies and their rationale are identified in electronic supplementary material, Pre-registration. All R code for data compilation, model generation, analysis and figures is publicly available and can be found on Github [58]. Our analysis focuses on a subset of individuals who were personally interviewed, one from each household (i.e. the focal interviewee), and on whom complete records were available which generated 9870 farmers and 12 204 non-farmers. For a complete description of how data were collected see electronic supplementary material, Methods: data.

### (a) Variables

The dependent variable for all models was a binomial response indicating whether or not an evacuee remained in western Finland (0) or returned to Karelia (1) between 1942 and 1944. Predictors for personal characteristics of the evacuees, geographical distance between the origin and destination location, sociocultural differences (e.g. linguistic differences) and ecological differences (e.g. average temperature) are shown in table 1 and described in more detail in electronic supplementary material, Results: predictor variables.

### (b) Statistical analysis

We used Bayesian inference for all statistical analyses (see electronic supplementary material, Bayesian priors and posterior distributions). To analyse the probability of an evacuee returning to their home municipality, we ran models on two subsets of evacuees—farmers ($n = 9870$) and non-famers ($n = 12 204$) (see above). We used the rstanarm package [59] in R Studio v. 3.3.3 [60] to run a Bayesian generalized linear mixed-effects model (GLMM) logistic regression (see electronic supplementary material, R code for all models).

To analyse how various sociocultural, ecological, geographical factors and personal characteristics of the evacuees themselves impacted the likelihood of returning to their birth municipality in Karelia, we first divided the sample into non-farmers and farmers. This was done because many of our ecological predictors (e.g. soil types) were only expected to affect the decisions of farmers. However, dividing our sample into non-farmers and farmers offered additional benefits, and allowed us to both cross validate [61] predictions generated on non-farmers on a new dataset of farmers and to determine which variables were unlikely to be associated with the skills and cultural adaptations of farmers in particular (see electronic supplementary material, Methods: validating predictors).

For **H1** (*Better Environment Hypothesis*), we entered the raw difference between the variables in the origin and destination (i.e. origin value − destination value) which includes the direction of this difference. For example, these models used the difference between the yearly rainfall in the evacuee's municipality of origin minus the yearly rainfall in their destination municipality. Because what constitutes a 'better' environment' is crop dependent (e.g. two of the most common crops in Finland

at that time, barley and rye depend oppositely on precipitation and temperature such that barley production depends more on high rainfall and rye depends more on warmer temperatures) [62], any significant directional preference (i.e. a 97.5% Credibility Interval that did not overlap with zero) which was only expressed by farmers was interpreted as potential evidence of a 'better' environment. However, some ecological factors, such as more rocks in the soil, are indicative of worse soil conditions for all farmers. It is also important to note, however, that these directional differences do not apply to the personal characteristics of the evacuees—age, sex, when and who they married—or to geographical and linguistic distance, which do not change depending on the point of origin (e.g. the distance from New York to Chicago is identical to the distance from Chicago to New York).

For **H2** (*Similar Environment Hypothesis*), we entered the positive distance between the variables in the origin and destination locations. For these models, the direction of this difference was not considered, and all differences were entered as the absolute value of the difference between the origin and destination municipality. Although linguistic distance cannot be interpreted as being better or worse, more similar dialects can be considered to constitute more similar socio-cultural environments. Just as the results from models using non-farmers can be used to illuminate which factors are unlikely to be specifically related to the skills of farmers, determining any directional preferences of evacuees in models testing **H1** can be leveraged to identify factors which are unlikely to be associated with cultural adaptations in models testing **H2** (see electronic supplementary material, Methods: model selection).

Data accessibility. The data and code used to generate these results and figures and that support the findings of this study are publicly available without restriction on Github: https://github.com/robertlynch66/Migrations_and_ecology [58].

Authors' contributions. R.L. wrote the first draft of the manuscript, conducted the statistical analysis and created the figures and tables. O.V. and T.H. conceived of the idea for the study and provided the ecological and linguistic data. J.L. and V.L. planned the study and oversaw data collection. J.P. did much of the historical research. All authors helped to write and edit the manuscript. R.F.L.: conceptualization, data curation, formal analysis, methodology, software, visualization, writing original draft, writing review & editing; J.L.: conceptualization, funding acquisition, project administration, supervision, writing review & editing; V.L.: conceptualization, funding acquisition, project administration, supervision, writing review & editing; T.H.: conceptualization, data curation, formal analysis, methodology, writing review & editing; J.P.: data curation, validation, writing review & editing; O.V.: conceptualization, project administration, resources, supervision, validation, writing original draft, writing review & editing.

Competing interests. We declare we have no competing interests.

Funding. No funding has been received for this article.

Acknowledgements. The Kone Foundation provided funding to R.L. and J.L. V.L. also acknowledges funding from The Academy of Finland. We thank Juuso Kallioniemi and Tuomas Salmi for helping to extract the data from the original texts.

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
