## [Peer Review File · Proceedings of the Royal Society B: Biological Sciences]

Review History

RSPB-2021-1014.R0 (Original submission)

Review form: Reviewer 1

Recommendation

Reject – article is not of sufficient interest (we will consider a transfer to another journal)

Scientific importance: Is the manuscript an original and important contribution to its field?
Acceptable

General interest: Is the paper of sufficient general interest?

Acceptable

Quality of the paper: Is the overall quality of the paper suitable?

Acceptable

Is the length of the paper justified?

Yes

Should the paper be seen by a specialist statistical reviewer?

Yes

Do you have any concerns about statistical analyses in this paper? If so, please specify them explicitly in your report.

No

It is a condition of publication that authors make their supporting data, code and materials available - either as supplementary material or hosted in an external repository. Please rate, if applicable, the supporting data on the following criteria.

Is it accessible?

Yes

Is it clear?

Yes

Is it adequate?

No

Do you have any ethical concerns with this paper?

No

Comments to the Author

There would be some general interest in the results of this paper, as it relates to migration studies and even applications to planning policy. Even though the paper is easily readable and the premise well-founded, the results should be restructured, and there needs to be greater consistency in nomenclature throughout to increase clarity.

Problematically, it is unclear that the results of this study would make any significant or novel contribution. There are other datasets available, or alternative research designs, that would better serve testing the two hypotheses the authors posit. Arguably, the authors stretch the application of the dataset used beyond its limitations. For instance, looking at the "Better Environments" hypothesis for farmers, the authors do not acknowledge the niche specialization of farming practices that are borne from specific ecological contexts, making it difficult to adapt this knowledge to novel environments. It could be argued that any change in this instance would result in negative outcomes and therefore greater failed migration. Adoption of new techniques (likely through social learning) would strongly influence success in the novel environment and reduce failed migration. Social learning is facilitated through strong social networks. Therefore, it is difficult to parse these variables and test the two hypotheses independently. I appreciate the use of existing data; however, selecting a more appropriate dataset (preferably one containing greater contextual or qualitative information that could be coded) or research design to answer these questions would be wise. Likewise, the impetus for migration can strongly influence behavioral outcomes, such as the likelihood of return migration. There is no discussion on push-pull dynamics of migration nor any references to migration psychology that have overt influence in forcibly displaced populations. It is acknowledged that this information cannot be extracted from the dataset used and likely is not the purpose of the paper; however, these limitations should be noted by the authors.

Data should also be included on return rates over the 4 year period from 1941 - 1944 where more than 50% of the displaced population returned to Karelia. What was the distribution in return over these years? Also, is there any understanding of whether the host sites were selected by displaced individuals, with preference for similar cultural groups as host / receiving communities? This has implications for geographic distribution of evacuated members, and could make for a greater discussion on geographic distance and return migration, which is more of a novel contribution.

The authors balance paper length well with adequate figures and tables provided. It would have been helpful to provide a map with geographic scale to visualize the population distribution,

along with a mention of the historic timeline and migration flows early in the text. Where appropriate, data labels on the x-axis should be included.

Review form: Reviewer 2

Recommendation

Major revision is needed (please make suggestions in comments)

Scientific importance: Is the manuscript an original and important contribution to its field?

Good

General interest: Is the paper of sufficient general interest?

Good

Quality of the paper: Is the overall quality of the paper suitable?

Good

Is the length of the paper justified?

Yes

Should the paper be seen by a specialist statistical reviewer?

Yes

Do you have any concerns about statistical analyses in this paper? If so, please specify them explicitly in your report.

No

It is a condition of publication that authors make their supporting data, code and materials available - either as supplementary material or hosted in an external repository. Please rate, if applicable, the supporting data on the following criteria.

Is it accessible?

Yes

Is it clear?

Yes

Is it adequate?

Yes

Do you have any ethical concerns with this paper?

No

Comments to the Author

See attached review. (See Appendix A)

Review form: Reviewer 3

Recommendation

Accept with minor revision (please list in comments)

Scientific importance: Is the manuscript an original and important contribution to its field?

Good

General interest: Is the paper of sufficient general interest?

Good

Quality of the paper: Is the overall quality of the paper suitable?

Good

Is the length of the paper justified?

Yes

Should the paper be seen by a specialist statistical reviewer?

No

Do you have any concerns about statistical analyses in this paper? If so, please specify them explicitly in your report.

No

It is a condition of publication that authors make their supporting data, code and materials available - either as supplementary material or hosted in an external repository. Please rate, if applicable, the supporting data on the following criteria.

Is it accessible?

Yes

Is it clear?

Yes

Is it adequate?

Yes

Do you have any ethical concerns with this paper?

No

Comments to the Author

I really enjoyed reading Lynch et al manuscript on the impacts of socio-cultural similarity on human migrations. The manuscript attempts to compare the support for two hypotheses related to human migrations: the better environment hypotheses (after relocation, individuals are more likely to remain in a place with a superior environment) and the similar environment hypothesis (after relocation, individuals are more likely to remain in a place more similar to their place of origin). Their results show that cultural, rather than environmental, factors more likely affect the success of a migration event (i.e., the probability of individuals to remain after a relocation event). The manuscript is very well written and the predictors, the potential implications and caveats are discussed at length. The authors take advantage of a rich dataset to shine light on human migration patterns. The results are interesting, and I think they will be of interest to quite a few disciplines (e.g. cultural evolution, linguistics, human history, sociology).

Introduction: I thought the introduction was a nice read and good background into the topic. I did notice a couple of typos:

- Line 35 is missing a bracket.
- Line 40: "an important driver of human dispersal"

Materials and methods: I generally found this section good and detailed. I did get confused at first by what the authors meant by when saying "farmers", and given the distinction between farmers and non-farmers analyses, I think this could be clarified more – i.e., at first I thought

farmers meant people relying on crops as their income source (and not livestock), but in discussion it was more clear that farmers meant a profession (i.e. separating farmers from people working in a factory for example).

Another question I had was about the financial/economics aspects of the two regions: I understand taxes are included, but what was the general financial prospect of individuals living in Karelia vs Western Finland? I understand data on income might not be available, but I was wondering if the authors can generally comment on this? It would make sense to me that living standards based on income or the general wealth of a Western Finland vs Karelia would also factor in the process of moving back to Karelia? This comment ties also with "Validating predictors" section – for example, %peat and moraine found in soil are predictive of the likelihood of non-farmers returning might not mean that the factors are not good predictors for the specific cultural skills and adaptation of farmers, but it could be that they provide an advantage to the farming sector that then translates into economic wealth for the whole region (which will indirectly also apply non-farmers). This also related to lines 286-291.

Few more specific comments:

- Line 531: is the question mark supposed to be there?
- Line 576: I might have gotten confused by the phrasing but moving east-west would mean similar latitudes not longitudes.
- Line 598: were there no unmarried individuals?
- Lines 641-645: are linguistic distances computed including only features that were present in both dialects? (in other words, how do authors deal with the absence of a certain feature when comparing a pair of municipal dialects?)
- Line 720: can the authors mention the percentage of 0 and 1 in the outcome variable in these models (just so the reader can assess if there is any imbalance in the models)?

Results: I thought the results section was well written, but I found the figures and table one confusing.

Regarding Fig 1 and Fig 2 – these are not referenced in the main text, and I honestly did not understand what is going on with them. I assumed they represent results from Table 1 (Better and Similar environment models), but I think it would be great if the figure legend would explain in words what the figures are showing. I based my review on table 1 and figures S5-S6.

Regarding Table 1: the two hypotheses mentioned through-out the manuscript were the similar and better environment hypotheses. Then Table 1 introduces the "Different environment" model (?). The supplemental material explains the different environment hypothesis as being the complement to the similar environment hypothesis, but this all should be explained in the main text. I was confused in general about this different environment top model – if I read this correctly, the top model is basically the reduced model following model selection criteria? But then following Table S2 (and lines 197-200) – wouldn't that be two models: a reduced model for better environment hypothesis and a reduced model for similar environment hypothesis?

Lines 284-285: I think there are some brackets missing here (closing bracket after better env hypothesis and closing bracket after similar env hypothesis).

Lines 300-303: I understand the rationale here, but these are still interesting results, and I think it would be worth expanding at least a little bit.

Discussion: I thought the discussion was very thorough and I enjoyed reading it. A few minor comments:

Lines 423-436: It might be just me, but I did not understand the rationale here.

Lines 474-488: I really appreciated this paragraph, as it echoed some concerns I had reading the paper. Regarding the last line about east-west vs north-south differences, I think it would help if the authors would provide details on how big the differences in environmental conditions (e.g. rainfall or temperature) are in the sample. It might be that these differences are simply not big enough to influence crops or livestock (i.e. the crops might be much more robust compared to the differences recorded in the data, and so it is not that environment does not matter, rather the scale of differences in the sample is too small). I think this is important to mention, as the lack of effect in ecological factors could be attributed to scale.

Decision letter (RSPB-2021-1014.R0)

21-Jun-2021

Dear Dr Lynch:

I am writing to inform you that your manuscript RSPB-2021-1014 entitled "Socio-cultural similarity with host population rather than ecological similarity predicts success and failure of human migrations" has, in its current form, been rejected for publication in Proceedings B.

This action has been taken on the advice of referees, who have recommended that substantial revisions are necessary. With this in mind we would be happy to consider a resubmission, provided the comments of the referees are fully addressed. However please note that this is not a provisional acceptance.

Sincerely,
Dr Robert Barton
<mailto:proceedingsb@royalsociety.org>

Associate Editor
Comments to Author:

This is an interesting paper exploiting a historical 'natural experiment' to test what factors predict successful human migrations. This addresses a set of key questions about an important, and difficult to study, aspect of human behavior that also has important implications for understanding human history. However, the reviewers also raise several points about the methods and interpretation of the study results that should be addressed. One common thread in the reviews, which I agree with, is that the specific analytical methods need to be better explained. This includes definitions of outcomes as well as better explanations concerning how different alternative pathways to migration success were controlled for. Given that this is necessarily an observational study these controls (e.g., accounting for variance in taxes / financial status across regions) are crucial for the paper to be relevant to a broad biological audience. A

second concerns the novelty of the data and more generally whether this dataset actually provides an appropriate test of the two main hypotheses (e.g. whether similar environments versus better environments predict success). This point is raised in a couple of ways. For example, R1 notes that farmers may not be the best test of these hypotheses given the degree to which farming practices are dependent on social learning to appropriately exploit. In that sense, there is some potential conflation between an environment being “similar” and it being “better” – as a given environment may in fact be worse from the perspective of the migrant if they do not have the appropriate knowledge or tools to exploit it. R2 similarly points to other datasets (or other uses of the current dataset) that may better address these questions.

Reviewer(s)' Comments to Author:

Referee: 1

Comments to the Author(s)

There would be some general interest in the results of this paper, as it relates to migration studies and even applications to planning policy. Even though the paper is easily readable and the premise well-founded, the results should be restructured, and there needs to be greater consistency in nomenclature throughout to increase clarity.

Problematically, it is unclear that the results of this study would make any significant or novel contribution. There are other datasets available, or alternative research designs, that would better serve testing the two hypotheses the authors posit. Arguably, the authors stretch the application of the dataset used beyond its limitations. For instance, looking at the "Better Environments" hypothesis for farmers, the authors do not acknowledge the niche specialization of farming practices that are borne from specific ecological contexts, making it difficult to adapt this knowledge to novel environments. It could be argued that any change in this instance would result in negative outcomes and therefore greater failed migration. Adoption of new techniques (likely through social learning) would strongly influence success in the novel environment and reduce failed migration. Social learning is facilitated through strong social networks. Therefore, it is difficult to parse these variables and test the two hypotheses independently. I appreciate the use of existing data; however, selecting a more appropriate dataset (preferably one containing greater contextual or qualitative information that could be coded) or research design to answer these questions would be wise. Likewise, the impetus for migration can strongly influence behavioral outcomes, such as the likelihood of return migration. There is no discussion on push-pull dynamics of migration nor any references to migration psychology that have overt influence in forcibly displaced populations. It is acknowledged that this information cannot be extracted from the dataset used and likely is not the purpose of the paper; however, these limitations should be noted by the authors.

Data should also be included on return rates over the 4 year period from 1941 - 1944 where more than 50% of the displaced population returned to Karelia. What was the distribution in return over these years? Also, is there any understanding of whether the host sites were selected by displaced individuals, with preference for similar cultural groups as host / receiving communities? This has implications for geographic distribution of evacuated members, and could make for a greater discussion on geographic distance and return migration, which is more of a novel contribution.

The authors balance paper length well with adequate figures and tables provided. It would have been helpful to provide a map with geographic scale to visualize the population distribution, along with a mention of the historic timeline and migration flows early in the text. Where appropriate, data labels on the x-axis should be included.

Referee: 2

Comments to the Author(s)

See attached review

Referee: 3

Comments to the Author(s)

I really enjoyed reading Lynch et al manuscript on the impacts of socio-cultural similarity on human migrations. The manuscript attempts to compare the support for two hypotheses related to human migrations: the better environment hypotheses (after relocation, individuals are more likely to remain in a place with a superior environment) and the similar environment hypothesis (after relocation, individuals are more likely to remain in a place more similar to their place of origin). Their results show that cultural, rather than environmental, factors more likely affect the success of a migration event (i.e., the probability of individuals to remain after a relocation event). The manuscript is very well written and the predictors, the potential implications and caveats are discussed at length. The authors take advantage of a rich dataset to shine light on human migration patterns. The results are interesting, and I think they will be of interest to quite a few disciplines (e.g. cultural evolution, linguistics, human history, sociology).

Introduction: I thought the introduction was a nice read and good background into the topic. I did notice a couple of typos:

- Line 35 is missing a bracket.
- Line 40: "an important driver of human dispersal"

Materials and methods: I generally found this section good and detailed. I did get confused at first by what the authors meant by when saying "farmers", and given the distinction between farmers and non-farmers analyses, I think this could be clarified more – i.e., at first I thought farmers meant people relying on crops as their income source (and not livestock), but in discussion it was more clear that farmers meant a profession (i.e. separating farmers from people working in a factory for example).

Another question I had was about the financial/economics aspects of the two regions: I understand taxes are included, but what was the general financial prospect of individuals living in Karelia vs Western Finland? I understand data on income might not be available, but I was wondering if the authors can generally comment on this? It would make sense to me that living standards based on income or the general wealth of a Western Finland vs Karelia would also factor in the process of moving back to Karelia? This comment ties also with "Validating predictors" section – for example, %peat and moraine found in soil are predictive of the likelihood of non-farmers returning might not mean that the factors are not good predictors for the specific cultural skills and adaptation of farmers, but it could be that they provide an advantage to the farming sector that then translates into economic wealth for the whole region (which will indirectly also apply non-farmers). This also related to lines 286-291.

Few more specific comments:

- Line 531: is the question mark supposed to be there?
- Line 576: I might have gotten confused by the phrasing but moving east-west would mean similar latitudes not longitudes.
- Line 598: were there no unmarried individuals?
- Lines 641-645: are linguistic distances computed including only features that were present in both dialects? (in other words, how do authors deal with the absence of a certain feature when comparing a pair of municipal dialects?)
- Line 720: can the authors mention the percentage of 0 and 1 in the outcome variable in these models (just so the reader can assess if there is any imbalance in the models)?

Results: I thought the results section was well written, but I found the figures and table one confusing.

Regarding Fig 1 and Fig 2 – these are not referenced in the main text, and I honestly did not understand what is going on with them. I assumed they represent results from Table 1 (Better and Similar environment models), but I think it would be great if the figure legend would explain in words what the figures are showing. I based my review on table 1 and figures S5-S6.

Regarding Table 1: the two hypotheses mentioned through-out the manuscript were the similar and better environment hypotheses. Then Table 1 introduces the "Different environment" model (?). The supplemental material explains the different environment hypothesis as being the

complement to the similar environment hypothesis, but this all should be explained in the main text. I was confused in general about this different environment top model – if I read this correctly, the top model is basically the reduced model following model selection criteria? But then following Table S2 (and lines 197-200) – wouldn't that be two models: a reduced model for better environment hypothesis and a reduced model for similar environment hypothesis?

Lines 284-285: I think there are some brackets missing here (closing bracket after better env hypothesis and closing bracket after similar env hypothesis).

Lines 300-303: I understand the rationale here, but these are still interesting results, and I think it would be worth expanding at least a little bit.

Discussion: I thought the discussion was very thorough and I enjoyed reading it. A few minor comments:

Lines 423-436: It might be just me, but I did not understand the rationale here.

Lines 474-488: I really appreciated this paragraph, as it echoed some concerns I had reading the paper. Regarding the last line about east-west vs north-south differences, I think it would help if the authors would provide details on how big the differences in environmental conditions (e.g. rainfall or temperature) are in the sample. It might be that these differences are simply not big enough to influence crops or livestock (i.e. the crops might be much more robust compared to the differences recorded in the data, and so it is not that environment does not matter, rather the scale of differences in the sample is too small). I think this is important to mention, as the lack of effect in ecological factors could be attributed to scale.

Author's Response to Decision Letter for (RSPB-2021-1014.R0)

See Appendix B.

RSPB-2021-2298.R0

Review form: Reviewer 1

Recommendation

Accept with minor revision (please list in comments)

Scientific importance: Is the manuscript an original and important contribution to its field?

Good

General interest: Is the paper of sufficient general interest?

Acceptable

Quality of the paper: Is the overall quality of the paper suitable?

Acceptable

Is the length of the paper justified?

Yes

Should the paper be seen by a specialist statistical reviewer?

Yes

Do you have any concerns about statistical analyses in this paper? If so, please specify them explicitly in your report.

No

It is a condition of publication that authors make their supporting data, code and materials available - either as supplementary material or hosted in an external repository. Please rate, if applicable, the supporting data on the following criteria.

Is it accessible?

Yes

Is it clear?

Yes

Is it adequate?

Yes

Do you have any ethical concerns with this paper?

No

Comments to the Author

The authors have adequately addressed the major concerns that I had in my initial review. There are a number of minor corrections that should be made for the sake of clarity, and there are two substantive issues that I would also like to raise.

1. The authors should check terminologies closely in the final review to be sure that word choice is standardized throughout. For instance, "environments," I believe is a general reference to both ecological and sociocultural environments. There are a few instances in the paper where it appears that authors use "environment" in specific reference to ecological parameters. Please clearly define the intention for the use of the word "environment" and please make clear when discussing natural environments (i.e., ecologies) or when referencing social environments. Any usage is acceptable, as long as it is clearly defined and used systematically throughout the text.
2. Authors should revisit lines 22-24; 82-83; 748-761 to address confusion in word choice / editing issues.
3. Lines 238 - 240 More systematic to reference % probability of remaining, to fall in line with previous statements in the text and how data are displayed in the relevant plots
4. Lines 736 - 747 Relocate some of this explanation to the beginning of the main text, as it is important for the reader to understand the definition of "better" environments, as explained here - particularly the agnosticism of what parameters constitute a "better" ecological condition.
5. Line 328, Table 1 H2 is listed as "Different Environments" in the table. Isn't H2 in reference to the Similar Environments Hypothesis? Make sure there is consistency throughout with terminology, as this is confusing.
6. Line 446 We cannot presume clay soils are worse for farming. Shallow roots and cruciferous vegetables prefer the moisture retention found in clay soils. Again, this muddles an understanding of what a "better" environment is for farmers when authors declare some ecological parameters as favourable compared to others.

More substantive comments:

1. Although the authors have provided adequate defense of their methodological design and interpretation of results to the extent of the scope of this paper, there is still general weakness in testing the Better Environments Hypothesis. We have no real understanding how the ecological parameters chosen (temperature, altitude, % lake, livestock, farmed land, rainfall, rock, moraine, peat) may bear (or not) on the behaviours of farmers post-relocation, or their adaptive capacity to modify landscapes that circumvent or mitigate any potentially less than optimal farming conditions, thus rendering the ecological differences between origin and destination as insignificant. Furthermore, because ecological data are gathered at the municipality level yet data are analysed at the individual level, it obscures individual variations in plots, where the

assumption is that farmers would be choosing to work parcels of land that are optimal within the surrounding areas.

In addition to this, the authors have now acknowledged this point; however the point still remains, that the better environments hypothesis presupposes that ecologies have little / no bearing on the outcomes of non-farmers. Although acknowledged, this is somehow still problematic in the analysis.

2. The second substantive issue is that there is a demographic problem with the data set. There is no mention in the text regarding variation in outcomes based on occupation and marital status vis-a-vis gender. Agricultural females are listed as "farmer housewife" in the dataset.

Presumably these individuals are married to male farmers who also appear in the same dataset? There may be a handful, but I could not locate any unmarried female farmers. There was a much greater likelihood that female non-farmers (secretaries, nurses, teachers, etc.) were unmarried compared to the female farmers. For the married female farmer population, the decision to return or remain would likely not be independent of her husband's decision to return or remain (as she is almost assuredly housewife to the farmer). Or do we know that all subjects are independent of one another (i.e., no subjects are married)? This would need to be addressed in the analysis.

Review form: Reviewer 2

Recommendation

Accept with minor revision (please list in comments)

Scientific importance: Is the manuscript an original and important contribution to its field?

Good

General interest: Is the paper of sufficient general interest?

Excellent

Quality of the paper: Is the overall quality of the paper suitable?

Excellent

Is the length of the paper justified?

Yes

Should the paper be seen by a specialist statistical reviewer?

No

Do you have any concerns about statistical analyses in this paper? If so, please specify them explicitly in your report.

No

It is a condition of publication that authors make their supporting data, code and materials available - either as supplementary material or hosted in an external repository. Please rate, if applicable, the supporting data on the following criteria.

Is it accessible?

Yes

Is it clear?

Yes

Is it adequate?

No

Do you have any ethical concerns with this paper?

No

Comments to the Author

See attached comments. (See Appendix C)

Decision letter (RSPB-2021-2298.R0)

12-Nov-2021

Dear Dr Lynch:

Your manuscript has now been peer reviewed and the reviews have been assessed by an Associate Editor. The reviewers' comments (not including confidential comments to the Editor) and the comments from the Associate Editor are included at the end of this email for your reference. As you will see, the reviewers and the Editors have raised some concerns with your manuscript and we would like to invite you to revise your manuscript to address them.

Research ethics:

Use of animals and field studies:

If your study uses animals please include details in the methods section of any approval and licences given to carry out the study and include full details of how animal welfare standards

were ensured. Field studies should be conducted in accordance with local legislation; please include details of the appropriate permission and licences that you obtained to carry out the field work.

It is a condition of publication that you make available the data and research materials supporting the results in the article (<https://royalsociety.org/journals/authors/author-guidelines/#data>). Datasets should be deposited in an appropriate publicly available repository and details of the associated accession number, link or DOI to the datasets must be included in the Data Accessibility section of the article (<https://royalsociety.org/journals/ethics-policies/data-sharing-mining/>). Reference(s) to datasets should also be included in the reference list of the article with DOIs (where available).

If you wish to submit your data to Dryad (<http://datadryad.org/>) and have not already done so you can submit your data via this link [http://datadryad.org/submit?journalID=RSPB&manu=\(Document not available\)](http://datadryad.org/submit?journalID=RSPB&manu=(Document%20not%20available)), which will take you to your unique entry in the Dryad repository.

Please submit a copy of your revised paper within three weeks. If we do not hear from you within this time your manuscript will be rejected. If you are unable to meet this deadline please let us know as soon as possible, as we may be able to grant a short extension.

Best wishes,

Dr Robert Barton

Associate Editor Board Member

Comments to Author:

This is a revision of a paper harnessing a 'natural experiment' to test predictors of human migrations. The reviewers and I agree that this is a responsive revision that has greatly improved the paper. However, there are several comments that should still be addressed that arose from

the new version of the manuscript. R2 raises two bigger points. The first concerns how ecological parameters relate to farming behavior. While this is likely not be something that can be directly addressed with new analyses, it could be discussed as a limitation of the current study that future work could address. The second comment concerns the demographics of the sample (e.g., were there married female and male farmers included in the sample that were treated as independent in analyses?) and this issue should be explicitly addressed either by confirming that this potential confound is not present in the data or by ensuring that analyses address this potential non-independence of migration decisions concerning married couples. R1 further indicates some relevant citations of relevant prior work that should be integrated, as well as some clarifications about data concerning land availability that should be addressed.

Reviewer(s)' Comments to Author:

Referee: 2

Comments to the Author(s).

see attached comments

Referee: 1

Comments to the Author(s).

The authors have adequately addressed the major concerns that I had in my initial review. There are a number of minor corrections that should be made for the sake of clarity, and there are two substantive issues that I would also like to raise.

1. The authors should check terminologies closely in the final review to be sure that word choice is standardized throughout. For instance, "environments," I believe is a general reference to both ecological and sociocultural environments. There are a few instances in the paper where it appears that authors use "environment" in specific reference to ecological parameters. Please clearly define the intention for the use of the word "environment" and please make clear when discussing natural environments (i.e., ecologies) or when referencing social environments. Any usage is acceptable, as long as it is clearly defined and used systematically throughout the text.
2. Authors should revisit lines 22-24; 82-83; 748-761 to address confusion in word choice / editing issues.
3. Lines 238 - 240 More systematic to reference % probability of remaining, to fall in line with previous statements in the text and how data are displayed in the relevant plots
4. Lines 736 - 747 Relocate some of this explanation to the beginning of the main text, as it is important for the reader to understand the definition of "better" environments, as explained here - particularly the agnosticism of what parameters constitute a "better" ecological condition.
5. Line 328, Table 1 H2 is listed as "Different Environments" in the table. Isn't H2 in reference to the Similar Environments Hypothesis? Make sure there is consistency throughout with terminology, as this is confusing.
6. Line 446 We cannot presume clay soils are worse for farming. Shallow roots and cruciferous vegetables prefer the moisture retention found in clay soils. Again, this muddles an understanding of what a "better" environment is for farmers when authors declare some ecological parameters as favourable compared to others.

More substantive comments:

1. Although the authors have provided adequate defense of their methodological design and interpretation of results to the extent of the scope of this paper, there is still general weakness in testing the Better Environments Hypothesis. We have no real understanding how the ecological parameters chosen (temperature, altitude, % lake, livestock, farmed land, rainfall, rock, moraine, peat) may bear (or not) on the behaviours of farmers post-relocation, or their adaptive capacity to modify landscapes that circumvent or mitigate any potentially less than optimal farming conditions, thus rendering the ecological differences between origin and destination as insignificant. Furthermore, because ecological data are gathered at the municipality level yet data are analysed at the individual level, it obscures individual variations in plots, where the assumption is that farmers would be choosing to work parcels of land that are optimal within the surrounding areas.

In addition to this, the authors have now acknowledged this point; however the point still remains, that the better environments hypothesis presupposes that ecologies have little / no bearing on the outcomes of non-farmers. Although acknowledged, this is somehow still problematic in the analysis.

2. The second substantive issue is that there is a demographic problem with the data set. There is no mention in the text regarding variation in outcomes based on occupation and marital status vis-a-vis gender. Agricultural females are listed as "farmer housewife" in the dataset.

Presumably these individuals are married to male farmers who also appear in the same dataset?

There may be a handful, but I could not locate any unmarried female farmers. There was a much greater likelihood that female non-farmers (secretaries, nurses, teachers, etc.) were unmarried compared to the female farmers. For the married female farmer population, the decision to return or remain would likely not be independent of her husband's decision to return or remain (as she is almost assuredly housewife to the farmer). Or do we know that all subjects are independent of one another (i.e., no subjects are married)? This would need to be addressed in the analysis.

Author's Response to Decision Letter for (RSPB-2021-2298.R0)

See Appendix D.

Decision letter (RSPB-2021-2298.R1)

02-Dec-2021

Dear Dr Lynch

I am pleased to inform you that your manuscript entitled "Socio-cultural similarity with host population rather than ecological similarity predicts success and failure of human migrations" has been accepted for publication in Proceedings B.

Data Accessibility section

Open Access

Paper charges

Sincerely,

Dr Robert Barton

Associate Editor:

Board Member

Comments to Author:

The authors have done a nice job on the revision and addressed all the remaining points from the last round of reviews.

Appendix A

Review SPB-2021-1014

This paper uses a natural experiment in Finland where entire rural communities were forced to migrate during the World War II but then were allowed to return home. Using an empirical approach, the authors test two main hypotheses of successful migration: *The Better Environment Hypothesis* (individuals will remain in environments that are superior to the place they left) and the *Similar Environment Hypothesis* (individuals will remain if environments are ecologically and culturally more similar to the place they left). The authors find some support for the *Similar Environment Hypothesis*, i.e., socio-cultural similarity predicts successful migration rather than ecological similarity.

This is a very interesting paper, with a complex set of analyses, and interesting results. Although it is well-written, it is not always easy to follow the analytical methods that are employed without some more clarifications. I think the paper is potentially important, and makes important claims about the factors of human migration. However, I have several concerns related to the novelty of the data/approach and the overall goal of the paper from an evolutionary point of view.

One of my major concern with this study is about the novelty of the data and the lack of reference to existing studies. Natural settings to study the impact of forced migration on farmer's livelihoods have already previously used in economics. For example, Bazzi et al. 2016 analyzed the Transmigration Program in Indonesia that relocated migrants from rural Java and Bali to new rural settlements on islands. They show – in line with your similar environment hypothesis – that villages that assigned migrants from regions with more similar agro-climatic endowments have higher rice productivity one to two decades after the resettlement.

Sarvimäki et al. (2020) even use the same dataset (or parts of it) to investigate the long-run effects of resettling the Finnish population during World War II. They find that forced migration increases the likelihood to leave agriculture, which then leads to a large increase in long-run income among the displaced rural population while forced migration decreases the income of the resettled urban population. Sarvimäki et al. (2020) also aim to understand the mechanisms behind these effects (see <http://www.aalto-econ.fi/sarvimaki/forced.pdf>).

My specific points regarding the above literature:

- 1) You claim that the dataset of the Karelia in Finland has never been used in the context of understanding migration (see p. 133-142). Please outline what your work contributes to the above literature, particularly compared to Sarvimäki et al. (2020), and acknowledge it. I suggest that you can add novelty by testing hypotheses of cultural/ecological similarity in the context of forced migration (see also comment below).
- 2) While reading the manuscript, I was also not able to understand how you define the “success” or “failure” of human migration. In your analysis you use a simple binary variable that indicates if migrants stay or go back. If people stay, you interpret this as migration success. From a biological perspective (and also for the readers of a biological science journal), it would be much more interesting to see if your hypotheses can explain variations in the level of fitness. Measures of the level of fitness can be income or wealth but also health, reproduction or farm output/livestock (the latter can be used for farmers). How does cultural/ecological similarity predict the level of fitness of migrants that stay and of those that go back to Karelia? See also the literature above. For me it is not clear how you define success in this context.

Methods:

On p. 3 in lines 206-210 you refer to literature that indicates important factors that affect decisions to migrate: age and sex. What about other demographic characteristics and factors such as income, number of children etc.? Is the data not available?

Wealth heterogeneity plays an important role in shaping migration flows, see Bazzi (2017). Thus, wealth heterogeneity is part of the socio-cultural differences. Yet, wealth and income are completely missing in your analysis (see also comments above). Can you add wealth heterogeneity at the municipality level and use it to predict migration? In this way you include an important sociocultural factor which may support your findings even more (i.e., that socio-cultural similarity predicts migration).

How do you control for unknown characteristics that are not observable at the municipality level? Empirical models that look at the same issue (see citations above) usually include regional fixed effects to remove the omitted variable bias. Can you briefly explain how deal with this issue in your model (I have read the validating predictors section but I am not convinced that the issue is addressed sufficiently)?

Related to the comments above, how do you account for pre-war characteristics in your analysis? For example, if a farmer of Karelia was relatively wealthy (e.g., he owned land, assets, a lot of livestock, etc.) then I assume he is much more likely to come back to Karelia compared to farmers that are less wealthy. Please say more how you address this issue (maybe you can put more information on the individual characteristics in the appendix).

Literature:

Recent empirical literature has shown the importance of ecological similarity and variation in human behavior. For example, a recent paper by Barsbai et al. (2021) shows that ecological factors appear to operate consistently around the world. They find that human behavior from one location matches that of animals (birds and mammals) found at another location with the same ecological characteristics (incl. variables for migratory distance and day range, see Figure 2 in their paper). This finding is in contrast to your results, i.e., socio-cultural similarity rather than ecological similarity predicts successful migration. You should include these latest findings in your discussion.

Conceptual:

I am a bit confused about the formulation of the *Better Environment Hypothesis* and *Similar Environment Hypothesis* on p. 2 lines 162 to 170. If I understood correctly, the former includes only ecological variables, while the latter includes both - ecological and cultural variables. Why are you not considering cultural variables in the first one as well? Individuals can move from regions with weaker institutions to regions with better institutions (e.g., better infrastructure, better school system, higher/lower taxes etc.). On p. 3 lines 273 you mention the example of higher and lower taxes. I suggest to formulate the hypothesis more clearly at the beginning of p. 2.

References:

- Barsbai, T., Lukas, D., & Pondorfer, A. (2021). Local convergence of behavior across species. *Science*, 371(6526), 292-295.
- Bazzi, S., Gaduh, A., Rothenberg, A. D., & Wong, M. (2016). Skill transferability, migration, and development: Evidence from population resettlement in Indonesia. *American Economic Review*, 106(9), 2658-98.

Bazzi, S. (2017). Wealth heterogeneity and the income elasticity of migration. *American Economic Journal: Applied Economics*, 9(2), 219-55.

Sarvimäki, M., Uusitalo, R., & Jäntti, M. (2019). Habit formation and the misallocation of labor: evidence from forced migrations. Available at SSRN 3361356

Appendix B

REVIEWER COMMENTS

Dear Dr Lynch:

I am writing to inform you that your manuscript RSPB-2021-1014 entitled "Socio-cultural similarity with host population rather than ecological similarity predicts success and failure of human migrations" has, in its current form, been rejected for publication in Proceedings B.

This action has been taken on the advice of referees, who have recommended that substantial revisions are necessary. With this in mind we would be happy to consider a resubmission, provided the comments of the referees are fully addressed. However please note that this is not a provisional acceptance.

- 1) A 'response to referees' document including details of how you have responded to the comments, and the adjustments you have made.*
- 2) A clean copy of the manuscript and one with 'tracked changes' indicating your 'response to referees' comments document.*
- 3) Line numbers in your main document.*
- 4) Data - please see our policies on data sharing to ensure that you are complying (<https://royalsociety.org/journals/authors/author-guidelines/#data>).*

To upload a resubmitted manuscript, log into <http://mc.manuscriptcentral.com/prsb> and enter your Author Centre, where you will find your manuscript title listed under "Manuscripts with

Decisions." Under "Actions," click on "Create a Resubmission." Please be sure to indicate in your cover letter that it is a resubmission, and supply the previous reference number.

Sincerely,

Dr Robert Barton

Associate Editor

Comments to Author:

This is an interesting paper exploiting a historical 'natural experiment' to test what factors predict successful human migrations. This addresses a set of key questions about an important, and difficult to study, aspect of human behavior that also has important implications for understanding human history. However, the reviewers also raise several points about the methods and interpretation of the study results that should be addressed. One common thread in the reviews, which I agree with, is that the specific analytical methods need to be better explained. This includes definitions of outcomes as well as better explanations concerning how different alternative pathways to migration success were controlled for. Given that this is necessarily an observational study, these controls (e.g., accounting for variance in taxes / financial status across regions) are crucial for the paper to be relevant to a broad biological audience.

Thank you for your encouraging words and providing us an opportunity to thoroughly revise our manuscript based on the reviews. Please see our response to reviewer #2 (e.g., adding income as a predictor), but also our response to Reviewer #1 as it pertains to explaining how we differentiate between the 'Better' and 'Similar' environment hypotheses. Reviewer and editor comments are black and italicized and our response are plain text and green. All references to changes in the manuscript text are quoted, single spaced and italicized.

Also please note that upon submitting the revised manuscript we were alerted to the fact that BOTH the original and resubmitted manuscript (with all the additional changes) exceeded the proceedings B journal limit of 10 pages. Therefore, we have moved the following sections to Supplementary Materials: In the Methods section we have moved 'Historical Background',

'Data', 'Data Normalization', 'Model Selection', 'Model validity, effects and specifications' and have moved the first paragraph of the Statistical Analysis subsection to a new SI section called 'Bayesian priors and posterior distributions' and the last paragraph of this subsection to the SI section 'Validating Predictors'; in the Results section we have moved 'Personal characteristics' and 'Predictor variables'; in the Discussion we have shaved off a section of the 8th paragraph and put it in a new SI section called 'Additional caveats'. All the new sections are now referenced in the main manuscript. In addition 500 words were removed from the Discussion and conclusion to avoid repetition and shorten the manuscript to fit within the limit (see tarck changes document).

A second concern is the novelty of the data and more generally whether this dataset actually provides an appropriate test of the two main hypotheses (e.g. whether similar environments versus better environments predict success). This point is raised in a couple of ways. For example, R1 notes that farmers may not be the best test of these hypotheses given the degree to which farming practices are dependent on social learning to appropriately exploit. In that sense, there is some potential conflation between an environment being "similar" and it being "better" – as a given environment may in fact be worse from the perspective of the migrant if they do not have the appropriate knowledge or tools to exploit it. R2 similarly points to other datasets (or other uses of the current dataset) that may better address these questions.

We address these important issues in our response to Reviewer #1.

Reviewer(s)' Comments to Author:

Referee: 1

Comments to the Author(s)

There would be some general interest in the results of this paper, as it relates to migration studies and even applications to planning policy. Even though the paper is easily readable and the premise well-founded, the results should be restructured, and there needs to be greater consistency in nomenclature throughout to increase clarity.

Problematically, it is unclear that the results of this study would make any significant or novel contribution. There are other datasets available, or alternative research designs, that would better serve testing the two hypotheses the authors posit. Arguably, the authors stretch the

application of the dataset used beyond its limitations. For instance, looking at the "Better Environments" hypothesis for farmers, the authors do not acknowledge the niche specialization of farming practices that are borne from specific ecological contexts, making it difficult to adapt this knowledge to novel environments. It could be argued that any change in this instance would result in negative outcomes and therefore greater failed migration. Adoption of new techniques (likely through social learning) would strongly influence success in the novel environment and reduce failed migration. Social learning is facilitated through strong social networks. Therefore, it is difficult to parse these variables and test the two hypotheses independently.

Yes, as this reviewer writes, farming practices are 'borne from specific ecological contexts making it difficult to adapt this knowledge to novel environments'. This is true and it is a good point and introduces the potential conflation of what constitutes a better environment vs what constitutes a similar environment. This is a problem of which we were aware and sought to resolve but clearly failed to fully explain in the manuscript. Distinguishing between these two related reasons for successful migrations (better vs. similar environments) was in fact the entire reason for doing the study. The focus on farmers is precisely for this reason— because agricultural production is closely tied to environmental conditions and because farming techniques are culturally transmitted.

Our model testing the 'better environment hypothesis' suggests that the **only** ecological factor for which only farmers express a directional preference is rainfall such that farmers seem to prefer less rain. The others either fail to show a significant directional preference or are not specific to farmers suggesting that they are not that important. So, let's say that farmers who are used to growing barley — which they have learned from other farmers (e.g., their parents or neighbors) — evacuate to an area with less rainfall. This is of course worse for growing barley. But it is better for growing the heartier rye plant which the local host population grows. The question is now whether the evacuee can adapt to this new, worse (i.e., less rainfall) environment, start growing rye, and therefore remain in the new location. Why might the farmer be able to remain in the new location despite what appear to be worse conditions? The reviewer argues that this makes it difficult to distinguish between whether a given farmer returned home because of the lack of rainfall or because they could not adapt to the new conditions. Yes, on the face of it, this is true, and we need to use the models to distinguish between these two possibilities. It is also likely, as the reviewer points out, that the farmer

initially learned how to grow barley through social learning which is the same process that will allow him to switch to growing rye and therefore succeed in the new environment. However, in the novel environment, he has to learn how to do this from his new neighbors. The only assumption we need to make to distinguish between the models (better vs similar) is that the farmers who cannot adapt to the poorer (less rainfall) conditions by growing rye and return to their farms in Karelia (presumably continuing to grow barley) are those farmers who are less able to learn the new techniques from their new neighbors who are adept at it. If this ability to learn, adapt and remain depends on cultural affinity with the host population, such as linguistic similarity and being married to someone in the host population, then we might infer that these factors are more important for their success than ecological factors such as rainfall or soil type. Indeed, this is exactly what we find; sociocultural factors are the best predictors of remaining in the evacuation destination. More generally, however, the translocated farmers experience something called 'range expansion' as they need to adapt to the new environment. Here we are not really focused on determining what evolutionary process is causing the cultural adaptation which would be a topic for future research.

How can we distinguish between an initial preference for less rainfall (which was also presumably culturally transmitted in the origin location) and an inability to adapt to more rainfall (also culturally transmitted)? The key is that the old neighbors who they learned from were from the same ethnic, linguistic and cultural group (i.e., all Karelians) whereas the new neighbors differ on these characteristics and importantly do so to varying degrees. By leveraging these differences, we are able to distinguish between an initial preference and an inability to adapt to new conditions in three ways. First, and arguably most importantly, socio-cultural factors which indicate a greater affinity with the host population predict a greater likelihood of remaining. This is interesting in and of itself and it is important to emphasize that these social-cultural factors cannot be a test of the 'better environment' hypothesis because there is no *better* or *worse* for linguistic distance or marrying into the host population. Second, none of the ecological factors that could be characterized as *better* or *worse* (except less rainfall) seem to have any effect on return migration. In addition, the ecological factors that are almost certainly indicative of worse soil conditions for ALL farmers such as more clay and more rocks in the soil do not seem to have any impact on decisions to remain. Yes, it is true that in other cases, such as temperature and the length of the growing season, 'better' is likely to be contingent upon how familiar they are with these conditions. We analyze this in model 2 by testing whether the absolute difference between the origin and destination conditions affects their decision to return. On the whole, results from model 2 indicate that similar ecological conditions do not matter either, except that

farmers do seem to have a preference for similar, (i.e., neither warmer nor colder) temperatures which further supports the idea that they prefer similar environments. Finally, the top models support the similar environment hypothesis.

Overall, we agree that farming practices are culturally transmitted and that this applies to one's familiarity with ecological conditions such as temperature or rainfall as well as cultural affinity with the host population. However, if farmers consistently preferred better conditions, such as warmer temperatures and better soils, we would expect to see these preferences play out in the decisions of farmers as compared to non-farmers to return. We do not. Instead, we see that socio-cultural factors have the biggest effects on everyone (farmers and non-farmers alike). It is important to note, however, that this has little bearing on the question of whether evacuees prefer ecological conditions that are similar to what they are used to and therefore familiar to them. However, aside from temperature there does not appear to be strong evidence that farmers prefer other similar ecological conditions. Although the similar environment models decidedly outperform the better environment hypothesis, farmers, like everyone else in the population, just seem to prefer moving to areas where the host population is similar to themselves. Of course, whether or not socio-cultural similarity with the host population impacts on their ability to adapt their farming techniques to adapt to the new conditions is speculative.

We realize that much of this was not explicitly discussed in the manuscript. We have therefore made the following changes to the manuscript to help clarify:

In the Introduction (lines 164-166 track changes version) we have added the sentence:

“In other words, the success of non-farmers depends more on sociocultural factors than ecological conditions such as soil conditions whereas the success of farmers depends on both ecological and socio-cultural factors”

In the Discussion we have added a section (lines 450-466 track changes version):

“It might be argued, however, that because farming practices result from specific ecological contexts and that these practices are often learned from neighbors (i.e., via social learning), that it is difficult to parse these two influences and determine which is more important — sociocultural similarity or more similar ecological conditions. In this interpretation what is considered ‘better’ may be simply what the farmers are more used to. We are, however, able to distinguish between these possibilities in two ways. First, in the old environment the farmer learned how to grow specific crops in specific conditions in a relatively homogeneous population (i.e., the same ethnicity, dialect, and culture) whereas in the new environment, he or she has to learn how to do this from his new neighbors who vary along these dimensions. We are therefore

able to leverage this variance to determine the impact of socio-cultural similarity on the likelihood of remaining in the evacuation destination. Second, none of the ecological factors that are almost certainly indicative of worse soil conditions for all farmers, such as more clay or rocks in the soil have any impact on decisions to remain. Although the top models support the similar environment hypothesis, overall, it is hard to argue that ecological conditions matter very much at all, regardless of whether they are classified as better or similar.”

In the Methods Section we added (lines 758-759 track changes version):

“However, some ecological factors, such as more clay and more rocks in the soil, are indicative of worse soil conditions for all farmers.”

And

(Lines 769-771 track changes version): *“Although linguistic distance cannot be interpreted as being either better or worse (see above), a more similar dialect can be considered to constitute a more similar socio-cultural environment.”*

I appreciate the use of existing data; however, selecting a more appropriate dataset (preferably one containing greater contextual or qualitative information that could be coded) or research design to answer these questions would be wise. Likewise, the impetus for migration can strongly influence behavioral outcomes, such as the likelihood of return migration. There is no discussion on push-pull dynamics of migration nor any references to migration psychology that have overt influence in forcibly displaced populations. It is acknowledged that this information cannot be extracted from the dataset used and likely is not the purpose of the paper; however, these limitations should be noted by the authors.

These are all very good points, and we agree. We have noted this in the discussion section lines 491-499 (track changes version):

“Another limitation of the current study and the generalizability of these results to all migrants involves the specific circumstances in which Karelians were forcibly displaced by an invading army. While this allows us to more easily interpret these results (everyone was forced to leave regardless of any desire to migrate), the involuntary nature of the displacement is certain to have had an impact on the psychological motivations of the evacuees to return (Tabor and Milfont 2011; Stefanovic et al. 2015). Future studies using qualitative data that could be systematically coded assessing reasons why individuals report returning or remaining would help to better understand and interpret these results.”

Data should also be included on return rates over the 4-year period from 1941 - 1944 where more than 50% of the displaced population returned to Karelia. What was the distribution in return over these years?

We have added the following text “(see SM table S3 for return rates by year)” (lines 576-577 track changes version) and the following table (pasted below) which including the return rates for the years to the Supplementary materials; Table S3.

	Number of farmers	Percent	Number of non-farmers	Percent
Remained	1885	19%	5530	45%
Returned in 1942	7300	74%	6007	49%
Returned in 1943	622	6%	534	4%
Returned in 1944	80	1%	113	1%

Table S3. Return rates of farmers and non farmers by year: 1942-1944

Also, is there any understanding of whether the host sites were selected by displaced individuals, with preference for similar cultural groups as host / receiving communities? This has implications for geographic distribution of evacuated members, and could make for a greater discussion on geographic distance and return migration, which is more of a novel contribution.

There is no evidence that evacuees were deliberately moved to areas to which they were matched for cultural similarity, although some may have chosen to go to areas with which they were more familiar. The surprise attack launched by the Soviet Union into Finland on November 30, 1939 — just months after World War 2 began — that ended a mere three months later in March of 1940 with the Moscow Peace Treaty ensured a chaotic evacuation. Within two weeks of the treaty the Finns had to evacuate the entire ceded territory ³. Unlike the 2nd and ultimately permanent evacuation following what came to be called the ‘Continuation War’ in 1944 which the Finns had time to prepare for, the first evacuation was extraordinarily swift, and the government plan was very disorganized. Indeed, one of the reasons that we decided to conduct this study was to

capitalize on this largely random displacement of a single ethnic population in such a short period of time.

In the Methods Section we added (Lines 570-573):

“The initial evacuation was seen as a temporary measure, and it was widely assumed that the evacuees would return to Karelia at the end of the war. It also had to be completed within two weeks of the Treaty which led to a highly disorganized and largely chaotic displacement of the Karelian population across Western Finland (Waris, 1952).”

The authors balance paper length well with adequate figures and tables provided. It would have been helpful to provide a map with geographic scale to visualize the population distribution, along with a mention of the historic timeline and migration flows early in the text. Where appropriate, data labels on the x-axis should be included.

We have included a map (pasted below) and added text referencing it *“(see Figure 1)”* (line 141 track changes version) to show the evacuees locations after the initial resettlement following the winter war in 1940 and then after the territory was recaptured in the Continuation war in 1943 when individuals had the opportunity to return to their homes in Karelia.

Referee: 2

This paper uses a natural experiment in Finland where entire rural communities were forced to migrate during World War II but then were allowed to return home. Using an empirical approach,

the authors test two main hypotheses of successful migration: The Better Environment Hypothesis (individuals will remain in environments that are superior to the place they left) and the Similar Environment Hypothesis (individuals will remain if environments are ecologically and culturally more similar to the place they left). The authors find some support for the Similar Environment Hypothesis, i.e., socio-cultural similarity predicts successful migration rather than ecological similarity.

This is a very interesting paper, with a complex set of analyses, and interesting results. Although it is well-written, it is not always easy to follow the analytical methods that are employed without some more clarifications. I think the paper is potentially important, and makes important claims about the factors of human migration. However, I have several concerns related to the novelty of the data/approach and the overall goal of the paper from an evolutionary point of view.

One of my major concerns with this study is about the novelty of the data and the lack of reference to existing studies. Natural settings to study the impact of forced migration on farmer's livelihoods have already previously used in economics. For example, Bazzi et al. 2016 analyzed the Transmigration Program in Indonesia that relocated migrants from rural Java and Bali to new rural settlements on islands. They show – in line with your similar environment hypothesis – that villages that assigned migrants from regions with more similar agro-climatic endowments have higher rice productivity one to two decades after the resettlement.

Thank you for this excellent reference. The Bazzi et. al. study supports the relationship between environmental similarity and agricultural productivity by showing that similar climate and soil conditions are associated with higher rice production is consistent with our results. Neglecting to cite and review this study was a major oversight which we have now rectified.

In the Introduction we have added (lines 94-99):

“Indeed, Bazzi et. al. (2016) showed that villages in Indonesia which were assigned a higher proportion of migrants who had experienced similar climate and soil conditions achieved greater rice production, indicating that similar agro-climatic conditions are important for their transferability. Consistent with predictions of cultural evolution, they also showed that interactions with the host population and social capital in the resettlement areas were important predictors of rice productivity.”

In the Discussion we have added (lines 376-378 track changes version):

“In the previously mentioned study of Indonesian migrants, a one standard deviation increase in linguistic similarity between hosts and migrants predicted an increase in rice production by 25.8 percent (Bazzi, et. Al., 2016).”

This study and those cited by Bazzi et. al., 2016 are of course very relevant to our research questions and results. However, we do not believe these studies diminish the novelty of our study and they differ in at least two important ways. First, unlike the Karelian evacuation, participation in the Indonesian Transmigration Program was almost entirely voluntary and second, the Indonesian migrant populations did not experience a massive opportunity for reverse migration, thus preventing them from studying their preferences of staying or leaving as we do in our study.

Sarvimäki et al. (2020) even use the same dataset (or parts of it) to investigate the long-run effects of resettling the Finnish population during World War II. They find that forced migration increases the likelihood to leave agriculture, which then leads to a large increase in long-run income among the displaced rural population while forced migration decreases the income of the resettled urban population. Sarvimäki et al. (2020) also aim to understand the mechanisms behind these effects (see <http://www.aalto-econ.fi/sarvimaki/forced.pdf>).

My specific points regarding the above literature:

- 1) You claim that the dataset of the Karelia in Finland has never been used in the context of understanding migration (see p. 133-142). Please outline what your work contributes to the above literature, particularly compared to Sarvimäki et al. (2020), and acknowledge it. I suggest that you can add novelty by testing hypotheses of cultural/ecological similarity in the context of forced migration (see also comment below).*

Thanks for letting us clarify the differences. Although Sarvimäki et al. (2020) do analyze the same population of Karelian evacuees, they use a different and much more limited dataset. There are several crucial differences between their study and ours. Perhaps most importantly, is the different questions we are asking. Sarvimäki et al. (2020) are, as the reviewer mentions, using forced migrations to understand how habits can restrain people from making optimal economic decisions, while we are interested in how sociocultural and ecological factors affect

the success of migrations. Other important differences are that 1) the Sarvimäki study uses very little data on reverse migration during the war (1941-1944) and no individual level data, 2) is primarily concerned with economic outcomes after the war that caused people to leave farming and therefore doesn't use any individual level data prior to 1950, 3) their measure of whether people left agriculture is 25 years after the war in 1970, and 3) sample size differences (e.g. our study begins by using the entire displaced Karelian population while Sarvimäki et al. links a 10% sample of the 1950 census to 1971 tax records. As an important part of their study, however, they do seek to understand why farmers left agriculture after the war which does overlap with our study aims. For instance, they use the amount of private land that was expropriated from the host population to measure potential discrimination and resentment against the evacuees as a candidate for why farmers may have left farming 25 years after the war ended and found that it did not seem to play much of a role. Although we did not measure discrimination directly, other important similarities exist between our studies. This is especially true when Sarvimäki et al. discuss cultural differences and the strength of social networks as possible reasons why farmers decided to leave agriculture. These have been included and acknowledged in the revised manuscript:

In the Discussion we have added (lines 499-514 track changes version):

"It should also be mentioned that another study that asked why so many farmers left agriculture in the decades after the war using a different dataset from the same population came to a different conclusion. Sarvimäki et al (2019) showed that habits were an important factor in preventing farmers from leaving agriculture, and that neither cultural differences (measured by geographic distance between origin and host locations), nor weakened social networks (measured as the ratio of geographic distance between evacuees before vs after the evacuation), contributed much to why evacuees abandoned farming. It is important to note, however, that even though we also included geographic distance in our models and found similar results, we did not consider it to be a sociocultural factor. Instead, we relied more on linguistic differences because previous research had shown that in Finland, geographic distance is a poor predictor of cultural similarity and does not predict linguistic differences or contacts between dialects at all (Honkola et. al., 2018). In addition, geographic distance was not even included in our top ranked models for farmers. Finally, the Sarvimäki et. al. (2019) measure of the strength of evacuee social networks was limited to relationships within the evacuee population whereas we were primarily interested in relationships between evacuees and the host population."

2) *While reading the manuscript, I was also not able to understand how you define the "success" or "failure" of human migration. In your analysis you use a simple binary variable that indicates if migrants stay or go back. If people stay, you interpret this as migration success. From a biological perspective (and also for the readers of a*

biological science journal), it would be much more interesting to see if your hypotheses can explain variations in the level of fitness. Measures of the level of fitness can be income or wealth but also health, reproduction or farm output/livestock (the latter can be used for farmers). How does cultural/ecological similarity predict the level of fitness of migrants that stay and of those that go back to Karelia? See also the literature above. For me it is not clear how you define success in this context.

We address this issue in another paper and have added this to the Discussion (lines 519-526 track changes version):

“It is also important to note that success can be measured in different ways. Here we define it as whether evacuees returned to Karelia when they were given the opportunity. In a previous paper, however, we showed a tradeoff between reproduction and social status such that evacuees who remained obtained better jobs but lower fertility than those who returned, and suggest that this is likely to be the result of a complex array of social factors involving stronger within group ‘bonding’ social connections between Karelians who returned and more between group ‘bridging’ connections amongst those who remained (Lynch, et. al. 2019)”

Methods:

On p. 3 in lines 206-210 you refer to literature that indicates important factors that affect decisions to migrate: age and sex. What about other demographic characteristics and factors such as income, number of children etc.? Is the data not available?

Unfortunately, we do not have individual data on income. We do, however, have aggregate data on per capita income for each municipality which we have added to the manuscript (see response to next comment below). We also have number of children but adding it to the models does not notably affect any of the other parameter estimates or model comparisons. Although including the number of children does not have any notable effect on any of the other parameter estimates (shown in Table 1), it does produce significant results in the two models (better environment and similar environment) using non-farmers. In both models which do use the farmers — the primary focus of the manuscript — the models without the number of children entered perform better in model comparisons. Therefore, overall, we think it is better for comparisons between models (i.e., farmers vs non-farmers) to keep them out of all the models. Nevertheless, we have added the significant effects of the children to the results for non-farmers to the results section.

In the Results section we have added (lines 223-233):

“The number of children that evacuees had at the time of the evacuation was also added as a candidate variable to these models. The only significant results that survived model comparisons (i.e., the models performed better when it was included) were for models analyzing non-farmers. In the better environment model, each additional child for non-farmers resulted in a 2.6% expected increase in the likelihood of returning and in the similar environment model it increased their chances of returning by 3%. Neither of the models that included farmers were improved by adding reproduction (i.e., not significant and models including it received no weight in model comparisons), however. Because the exclusion of the number of children at the time of evacuation in any of the four models did not notably impact any of the other parameter estimates, it was not included in the final models.”

In the Methods section we have added (line 623 track changes version)”

“number of children at the time of evacuation”

Wealth heterogeneity plays an important role in shaping migration flows, see Bazzi (2017). Thus, wealth heterogeneity is part of the socio-cultural differences. Yet, wealth and income are completely missing in your analysis (see also comments above). Can you add wealth heterogeneity at the municipality level and use it to predict migration? In this way you include an important sociocultural factor which may support your findings even more (i.e., that socio-cultural similarity predicts migration).

We do have data on per capita income for each municipality as assessed in 1935. This measure, however, is highly correlated with taxes. Across municipalities in Karelia the correlation is 0.93 and across municipalities in western Finland it is 0.87. This led to Variance Inflation Factors (VIF's) for per capita income of 17.1 and 11.4 when we included it along with taxes for the better environment models using non-farmers and farmers, respectively. Due to this level of multicollinearity, we could not enter per capita income and taxes as predictors in the same models. We therefore ran all the models with each of them separately (see below) and with largely the same results. We have made the following changes to the manuscript and the Supplementary Materials:

In the Introduction we have added (lines 114-115 track changes version):

“However, because tax rates are strongly tied to income, we might also expect individuals to prefer areas with higher taxes and therefore higher productivity.”

In Results we have edited lines 267-269 in the track changes version to read:

“Municipalities with lower taxes and lower per capita income (see Table 1 and SM: Table S4, respectively) seem to be preferred by non-farmers and farmers alike [H1] so these preferences are unlikely to be specific to the needs of farmers.”

We reran all the models and model comparisons using income per capita instead of taxes and have created a new table which we have added to the Supplementary materials section as Table S4 (pasted below):

DV	Predictor	Factor type	Better Environment		Similar Environment		Top Model: [Similar Environment]			
			2.5% HDI	97.5% HDI	2.5% HDI	97.5% HDI	2.5% HDI	97.5% HDI		
RETURNED TO KARELIA	Intercept		-0.78	0.51	0.07	0.65	-0.08	0.44		
	Age	Personal characteristics	1.00	1.50	0.98	1.44	0.97	1.41		
	Sex	Personal characteristics	-0.47	-0.31	-0.43	-0.29	-0.44	-0.27		
	Married Before Evacuation	Personal characteristics	-0.01	0.11	-0.08	0.14	—	—		
	Geographic Distance	Geographic	0.07	0.74	0.43	1.15	0.25	1.09		
	Linguistic Distance	Sociocultural	0.66	1.26	0.43	1.02	0.48	0.97		
	Married out	Sociocultural	-0.40	-0.20	-0.40	-0.20	-0.41	-0.23		
	Married out X									
	Married before evacuation	Sociocultural	-0.30	0.15	-0.21	0.11	—	—		
	Education*	Sociocultural	-0.57	-0.34	-0.51	-0.28	-0.50	-0.32		
	Per Capita Income**	Sociocultural	-1.96	-1.40	-1.04	-0.62	-1.08	-0.71		
	Population density	Sociocultural	-0.21	0.75	-0.39	0.18	-0.36	0.38		
	NON-FARMERS (N= 12,204)	Specific to the livelihood of farmers	Mean Temperature	Ecological	-1.48	-0.49	-0.57	0.22	-0.44	0.08
			Median Altitude	Ecological	-1.56	-0.46	-0.25	0.36	—	—
Lake %			Ecological	0.42	1.02	-0.33	0.06	—	—	
Livestock no.			Ecological	-0.22	0.50	0.06	0.69	0.09	0.56	
Farmed %			Ecological	0.31	1.27	-0.35	0.14	—	—	
Rainfall			Ecological	-0.44	0.08	-1.51	-0.83	-1.48	-0.99	
Clay%			Ecological /Soil type	-0.06	0.46	-0.37	0.08	—	—	
Rock %			Ecological /Soil type	-0.15	0.46	-0.30	0.20	—	—	
Moraine %			Ecological /Soil type	0.80	1.60	-1.03	-0.56	-0.99	-0.64	
Peat %			Ecological /Soil type	-0.09	0.83	0.14	0.79	0.22	0.78	

DV	Predictor	Factor type	Better Environment		Similar Environment		Top Model: [Similar Environment]			
			2.5% HDI	97.5% HDI	2.5% HDI	97.5% HDI	2.5% HDI	97.5% HDI		
RETURNED TO KARELIA	Intercept		-0.50	1.60	0.27	1.23	0.23	1.18		
	Age	Personal characteristics	1.65	2.25	1.55	2.08	1.55	2.13		
	Sex	Personal characteristics	-0.25	-0.02	-0.25	0.00	-0.25	-0.04		
	Married Before Evacuation	Personal characteristics	0.10	0.42	0.07	0.40	0.12	0.39		
	Geographic Distance	Geographic	0.14	1.53	0.31	1.91	—	—		
	Linguistic distance	Sociocultural	0.26	1.38	0.19	1.36	1.03	1.58		
	Married out	Sociocultural	-0.82	-0.52	-0.83	-0.56	-0.80	-0.55		
	Married out X									
	Married before evacuation	Sociocultural	-0.43	0.38	-0.32	0.47	—	—		
	Education*	Sociocultural	—	—	—	—	—	—		
	Per Capita Income**	Sociocultural	-2.57	-1.43	-1.44	-0.51	-1.42	-0.68		
	Population density	Sociocultural	-0.84	1.06	-0.65	1.15	-0.65	0.77		
	FARMERS (N= 4,870)	Specific to the livelihood of farmers	Mean Temperature	Ecological	-1.71	0.10	0.28	1.77	0.65	1.99
			Median Altitude	Ecological	-1.72	0.28	-2.01	-0.39	-2.20	-0.65
Lake %			Ecological	0.67	1.48	-0.11	0.59	—	—	
Livestock no.			Ecological	-0.28	1.01	-0.06	1.05	-0.23	0.61	
Farmed %			Ecological	-0.12	1.77	-0.79	0.09	—	—	
Rainfall			Ecological	-1.08	-0.09	-1.30	-0.29	-1.42	-0.39	
Clay%			Ecological /Soil type	-0.54	0.59	-0.48	0.28	—	—	
Rock %			Ecological /Soil type	-0.42	0.61	-0.35	0.25	-0.38	0.31	
Moraine %			Ecological /Soil type	0.54	1.66	-0.55	0.18	-0.57	0.15	
Peat %			Ecological /Soil type	0.22	1.92	-0.75	0.19	—	—	

Table S4: Parameter estimates, Highest Density Intervals (HDI's) and Odds ratios for factors affecting the likelihood of returning to Karelia for non-farmers (top panel) and farmers (bottom panel) for the *better environment hypothesis* (left side) and the *similar environment hypothesis* (right side). Geographic distance and sociocultural factors (i.e. linguistic similarity, marrying outside of ones group both before and after the war and education for non-farmers) are the best predictors of remaining in the evacuation destination while ecological factors such as soil types, rainfall do not consistently predict the likelihood of reverse migration. Parameter estimates are in bold if 95% HDI does not overlap with zero and are in the predicted direction (i.e. positive estimates for models testing the *similar environment hypothesis*).

* Education only entered for models using non-farmers.

** Per capita income and taxes were highly correlated and therefore could not be entered into the same models (see Table 1 for models using taxes instead of per capita income).

In the Discussion we have edited line 439 in the track changes version to read:

“either lower taxes or lower per capita income”

And lines 443-445 (track changes version):

“Taxes and per capita income are likely to be directional effects (i.e., better environment hypothesis [H1]) and both lower taxes seem to be preferred by farmers and non-farmers alike.”

In the Methods section we have edited lines 673-674 in the track changes version to read:

“and per capita income was calculated as the mean household income of a given municipality in 1935.”

And in the Methods: Statistical Analysis section (lines 783-788 track changes version) we have added:

“Using a VIF of greater than 5 as a cutoff (Sheather 2009), we excluded per capita income which had a VIF of 17.1, 11.4, 7.2 and 8.4, respectively for each model when it was included together with taxes (see Supplementary materials: Table S1 for VIF’s for all predictors). The raw correlation between taxes and per capita income was also high (i.e., across municipalities in Karelia it was 0.93 and across municipalities in western Finland it was 0.87).”

Table S1 in Supplementary materials has been edited to include VIF’s for predictors for the models using the non-farmers (pasted here):

Predictors	VIF (farmers)	VIF (non- farmers)
1. age	1.2	1.2
2. sex	1.1	1.1
3. linguistic dist	2.5	2.3
4. geographic dist	2.4	2.3
5. outbred	1.3	1.3
6. married before war	1.3	1.6
7. temperature	3.0	4.1
8. rainfall	1.3	1.5
9. clay	2.5	2.7
10. altitude	2.9	3.8
11. lakes	1.3	1.5
12. moraine	1.9	2.2
13. peat	1.6	1.6
14. rock	1.9	2.0
15. farmed	4.1	3.8
16. livestock	1.8	1.9
17. density	1.3	1.3
18. taxes	2.4	3.9
19. outbred X married before war	1.2	1.5

Table S1. Variance Inflation Factors: We first selected variables to include in the model by eliminating all variables which were correlated and evaluated variance inflation factors for all predictors entered into the models. We selected predictors that were not correlated with each other and which had Variance Inflation Factors of less than 5.

How do you control for unknown characteristics that are not observable at the municipality level? Empirical models that look at the same issue (see citations above) usually include regional fixed effects to remove the omitted variable bias. Can you briefly explain how you deal with this issue in your model (I have read the validating predictors section but I am not convinced that the issue is addressed sufficiently)?

Yes, the possibility of omitted variable bias is always a problem in observational studies of this kind where data are limited, and we have attempted to both acknowledge and deal with this in ways that we describe in the *Validating Predictors* subsection. Birth municipality was included as a random effect in all the models to help with this issue but using the non-farmers as a kind of training data as described should also help with this issue. We also hope that by adding results which substitute taxes with per capita income in the revision (see above and SM Table S4) will help with this issue. Still, this is always a possibility with this type of research, and we have therefore added the following sentence to the Methods: *Validating Predictors* subsection (lines 855-859 track changes version) to further acknowledge this possibility:

“Even though ecological factors which affect non-farmers are unlikely to be good predictors of the specific cultural skills of farmers, it is important to note that they could indirectly influence the livelihoods of both farmers and non-farmers alike, by for example, generating economic wealth for the entire region which will also impact on the decision of non-farmers to return or remain.”

We also have added the following to the Discussion (lines 492-500 track changes version) to acknowledge that understanding the psychological motivations of evacuees could be better addressed by coding qualitative data asking them why they remained or returned which unfortunately we did not have for these individuals:

“Another limitation of the current study and the generalizability of these results to all migrants involves the specific circumstances in which Karelians were forcibly displaced by an invading army. While this allows us to more easily interpret these results (everyone was forced to leave regardless of any desire to migrate), the involuntary nature of the displacement is certain to have had an impact on the psychological motivations of the evacuees to return^{1,2}. Future studies using qualitative data that could be systematically coded assessing reasons why individuals report returning or remaining would help to better understand and interpret these results.”

Related to the comments above, how do you account for pre-war characteristics in your analysis? For example, if a farmer of Karelia was relatively wealthy (e.g., he owned land, assets, a lot of livestock, etc.) then I assume he is much more likely to come back to Karelia

compared to farmers that are less wealthy. Please say more how you address this issue (maybe you can put more information on the individual characteristics in the appendix).

Yes, this is a good point and one we did attempt to analyze but neglected to include in the final draft. We do have data on whether they owned land in Karelia but only had data on how much land they owned in 1970 and did not have a good measure of how much they owned prior to the evacuation. We have added the following to the methods section to address this issue and indicate why we do not believe that this issue would have an important effect on our results (lines 625-631 track changes version):

“We also coded and entered whether or not farmers owned any land prior to evacuation, but adding this dummy variable to the models did not substantially affect our results and was eliminated in model comparisons (see Model selection below), so was not included in any of the final models. Although we did not have data on how much land individuals owned prior to the evacuation, the amount of land farmers received was commensurate with their holdings in Karelia (Loehr et al. 2017), so failing to add this as a covariate is unlikely to have a major impact on our results.”

Literature:

Recent empirical literature has shown the importance of ecological similarity and variation in human behavior. For example, a recent paper by Barsbai et al. (2021) shows that ecological factors appear to operate consistently around the world. They find that human behavior from one location matches that of animals (birds and mammals) found at another location with the same ecological characteristics (incl. variables for migratory distance and day range, see Figure 2 in their paper). This finding is in contrast to your results, i.e., socio-cultural similarity rather than ecological similarity predicts successful migration. You should include these latest findings in your discussion.

The Barsbai paper is relevant to our results, and we have added the reference. Thank you. Why these results differ from our own could result from any number of factors. Mainly, however, they are assessing ecological variables in isolation and do not compare them to any sociocultural variable as this is not possible using birds and mammals. Second, they are broadly analyzing behavioral similarity across species living in the same areas and are not assessing migration success. We do not doubt that ecological factors have shaped human behavior and discuss this in the manuscript. However, we have now added the following text to the revised manuscript.

In the Introduction we have added (lines 71-74 track changes version):

“The importance of ecological conditions may also extend across species. Barsbai (2021), for example, showed that foraging, reproductive and social behaviors of humans, birds and mammals living in the same areas are remarkably similar.”

We have also added the following text to the Discussion (lines 515-519 track changes version):

“This is of course not to say that ecological factors do not matter at all, and broad comparisons between the behavior of humans and that of birds and mammals suggests that ecological conditions have important effects (Barsbai et al. 2021). However, in contexts such as the Karelian evacuation (e.g., where migrations cross relatively short distances and similar latitudes) these factors may be overridden by the importance of socio-cultural similarities.”

Conceptual:

I am a bit confused about the formulation of the Better Environment Hypothesis and Similar Environment Hypothesis on p. 2 lines 162 to 170. If I understood correctly, the former includes only ecological variables, while the latter includes both - ecological and cultural variables. Why are you not considering cultural variables in the first one as well? Individuals can move from regions with weaker institutions to regions with better institutions (e.g., better infrastructure, better school system, higher/lower taxes etc.).

Yes, this was confusing. The socio-cultural variables were included in models testing both hypotheses — the Better Environment Hypothesis and Similar Environment Hypothesis (see Table 1). But we realize now that the in lines 162-170 phrasing made this unnecessarily confusing. The distinction we want to make is that several of the sociocultural factors — ingroup marriages, dialect differences and education — can neither be interpreted as being ‘better’ or ‘worse’ and more practically could not be entered as ‘directional’ differences (e.g., dialects are merely different and cannot be higher or lower, better or worse). The sociocultural variables that could be entered into the respective models as both directional and absolute — taxes and population density— were entered both ways into each respective model. Although all the variables were entered into both models, the distinction we wanted to make was for HOW they were entered to help with our interpretation of the results.

Although we have mentioned this in **Results: Sociocultural factors** subsection we have edited these lines (lines 261-266 track changes version),

“It is important to note, however, that all of these sociocultural factors mentioned above reflect absolute distance because they cannot easily be characterized as being better or worse (e.g., ingroup marriages, dialect differences and education). Tax rate and population density, however, do reflect directional differences because they can be either lower or higher in each location.

And

“Municipalities with lower taxes and lower per capita income (see Table 1 and SM: Table S4, respectively) both seem to be preferred by non-farmers and farmers alike [H1] so these preferences are unlikely to be specific to the needs of farmers. Finally, population density, which also might be predicted to impact the decisions of all evacuees, does not seem to influence returning for either farmers or non-farmers in any of the models.”

“(lines 267-271 track changes version)”

We have also edited lines 170-176 (track changes version) to read:

“Specifically we predict that 1) individuals who move to environments that are superior to the environments in which they were living in Karelia (e.g. more rainfall, better soil or longer growing season) will be more likely to remain in their relocation municipality — The Better Environment Hypothesis or 2) that individuals who move to environments that are more similar to their origin location in Karelia will be more likely to remain in their relocation municipality — The Similar Environment Hypothesis.”

On p. 3 lines 273 you mention the example of higher and lower taxes. I suggest to formulate the hypothesis more clearly at the beginning of p. 2.

Yes, clarifying the distinction between directional and absolute differences and hence preferences is important. We have added the following to the top of page 2 (lines 112-115 in the track changes document):

“Other sociocultural and economic factors like tax rates, however, are more likely to elicit directional preferences with individuals preferring lower taxes (Liebig et al. 2007).”

References:

Barsbai, T., Lukas, D., & Ponderfer, A. (2021). Local convergence of behavior across species. Science, 371(6526), 292-295.

Bazzi, S., Gaduh, A., Rothenberg, A. D., & Wong, M. (2016). Skill transferability, migration, and development: Evidence from population resettlement in Indonesia. American Economic Review, 106(9), 2658-98.

Bazzi, S. (2017). Wealth heterogeneity and the income elasticity of migration. American Economic Journal: Applied Economics, 9(2), 219-55.

Sarvimäki, M., Uusitalo, R., & Jäntti, M. (2019). Habit formation and the misallocation of labor: evidence from forced migrations. Available at SSRN 3361356

Referee: 3

Comments to the Author(s)

I really enjoyed reading Lynch et al manuscript on the impacts of socio-cultural similarity on human migrations. The manuscript attempts to compare the support for two hypotheses related to human migrations: the better environment hypotheses (after relocation, individuals are more likely to remain in a place with a superior environment) and the similar environment hypothesis (after relocation, individuals are more likely to remain in a place more similar to their place of origin). Their results show that cultural, rather than environmental, factors more likely affect the success of a migration event (i.e., the probability of individuals to remain after a relocation event). The manuscript is very well written and the predictors, the potential implications and caveats are discussed at length. The authors take advantage of a rich dataset to shine light on human migration patterns. The results are interesting, and I think they will be of interest to quite a few disciplines (e.g. cultural evolution, linguistics, human history, sociology).

Introduction: I thought the introduction was a nice read and good background into the topic. I did notice a couple of typos:

- Line 35 is missing a bracket.

Fixed.

- Line 40: "an important driver of human dispersal"

Fixed.

Materials and methods: I generally found this section good and detailed. I did get confused at first by what the authors meant by when saying "farmers", and given the distinction between farmers and non-farmers analyses, I think this could be clarified more – i.e., at first I thought farmers meant people relying on crops as their income source (and not livestock), but in discussion it was more clear that farmers meant a profession (i.e. separating farmers from people working in a factory for example).

Yes, this is correct. We did have data on whether farmers owned land but only had data on how much land they owned in 1970 and we did not have a good measure of how much they owned prior to the evacuation. We have therefore added the following to the Methods section to clarify this issue (lines 625-632 track changes version):

“We also coded and entered whether or not farmers owned any land but adding this dummy variable to the models did not substantially affect our results and was eliminated in model comparisons (see Model selection below), so was not included in any of the final models. Although we did not have data on how much land individuals owned prior to the evacuation, the amount of land farmers received was commensurate with their holdings in Karelia (Loehr et al. 2017), so failing to add this as a covariate is unlikely to have a major impact on our results.”

Another question I had was about the financial/economics aspects of the two regions: I understand taxes are included, but what was the general financial prospect of individuals living in Karelia vs Western Finland? I understand data on income might not be available, but I was wondering if the authors can generally comment on this? It would make sense to me that living standards based on income or the general wealth of a Western Finland vs Karelia would also factor in the process of moving back to Karelia? This comment ties also with “Validating predictors” section – for example, %peat and moraine found in soil are predictive of the likelihood of non-farmers returning might not mean that the factors are not good predictors for the specific cultural skills and adaptation of farmers, but it could be that they provide an advantage to the farming sector that then translates into economic wealth for the whole region (which will indirectly also apply non-farmers). This also related to lines 286-291.

This is a very good point. We think that our section on validating predictors helps with some of these issues relating to the impact of wealth on the likelihood of returning to Karelia. Although we do not have data on income of the evacuees, we do have their occupation from which we might be able to approximate their socioeconomic status. Although education was not available for farmers because occupation was the variable that we used to assess education, the likelihood of returning was higher amongst more educated non-farmers. This also means that we are unable to differentiate SES amongst farmers except by categorizing them by whether they owned land (see response to comment directly above including changes to manuscript). In a previous paper, however, we did assess the effect of socioeconomic status which we also assessed by occupation. Findings from that analysis (Lynch, et. al., 2019) suggest that individuals who remained in western Finland (i.e. did not return) were more likely to marry into a higher social class (i.e. occupation) and that higher social classes were more likely to remain (i.e. technical professions and office workers). We also added per capita income to all the models (see Supplementary Materials Table S4) in the revision in response to a comment from another reviewer. The high correlation between taxes and per capita income, however, did not allow

us to enter taxes and per capita income into the same models (see response to reviewer #2 above and associated changes to the manuscript).

We have added to Results: *Ecological conditions* (lines 282-284 track changes version):

“It is however important to recognize that these variables may have indirect effects on the entire region which may then act upon farmers and non-farmers similarly. (see Methods: Validating predictors).”

We have also added the following to the Methods section: *Validating predictors* (lines 856-860):

“Even though ecological factors which affect non-farmers are unlikely to be good predictors of the specific cultural skills of farmers, it is important to note that they could indirectly influence the livelihoods of both farmers and non-farmers alike, by for example, generating economic wealth for the entire region which will also impact on the decision of non-farmers to return or remain.”

Few more specific comments:

- Line 531: is the question mark supposed to be there?

No, thank you. It was caused by an incomplete reference in Github. This has been fixed.

- Line 576: I might have gotten confused by the phrasing but moving east-west would mean similar latitudes not longitudes.

Yes. Of course. We have fixed this. Thank you.

- Line 598: were there no unmarried individuals?

No. There must have been some, but we do not have data on all wedding years which we used to determine this variable. Therefore, we have changed the text in lines 624-625 track changes version to read:

“or never married”

- Lines 641-645: are linguistic distances computed including only features that were present in both dialects? (in other words, how do authors deal with the absence of a certain feature

when comparing a pair of municipal dialects?)

Yes, that is correct. The distances were computed only using features that were present in both dialects. The linguistic data were obtained from the Dialect Atlas of Finnish which were collected by Lauri Kettunen in the 1920s and 1930s (Kettunen, 1940). The Atlas contains various linguistic features (phonological, morphological and lexical) and their variants in 525 Finnish-speaking municipalities. Within each municipality, Kettunen recorded the linguistic variants of 1–4 informants. Therefore, each municipality had at a minimum one variant and at maximum four variants of each linguistic feature.

- Line 720: can the authors mention the percentage of 0 and 1 in the outcome variable in these models (just so the reader can assess if there is any imbalance in the models)?

Yes, we have added these numbers “(n=9,870)” for farmers and “(n=12,204)” for non-farmers to the Methods is line 723 (track changes version).

Results: I thought the results section was well written, but I found the figures and table one confusing. Regarding Fig 1 and Fig 2 – these are not referenced in the main text, and I honestly did not understand what is going on with them. I assumed they represent results from Table 1 (Better and Similar environment models), but I think it would be great if the figure legend would explain in words what the figures are showing. I based my review on table 1 and figures S5-S6.

Thanks for noticing this. This was a big oversight. We have added the following text to the captions of what are after the revisions Figure 2:

“Closer geographic distance between the origin and destination locations, greater linguistic similarity, marrying someone from the host location (i.e., western Finland) and having a job that requires an education all strongly predict the likelihood of a non-farmer evacuee remaining in the host location. Model generated posterior distribution predictions (dark lines); credibility intervals (blue shading) drawn from the top model in model comparisons (see Supplementary Materials: Figure S5 for posterior distributions for all covariates in the model). The observed data (means and standard errors) are also shown with samples less than 50 removed.”

And have changed the caption of what is now Figure 3 to read:

“Greater linguistic similarity, marrying someone from the host location (i.e., western Finland) and similar temperatures between the origin and destination locations all strongly predict the likelihood of a farmer evacuee remaining in the host location. Model generated posterior distribution predictions (dark lines), credibility intervals (blue shading) drawn from the top model in model comparisons (see Supplementary Materials: Figure S5 for posterior distributions for all covariates in the model). The observed data (means and standard errors) are also shown with samples less than 50 removed.”

Regarding Table 1: the two hypotheses mentioned through-out the manuscript were the similar and better environment hypotheses. Then Table 1 introduces the “Different environment” model (?). The supplemental material explains the different environment hypothesis as being the complement to the similar environment hypothesis, but this all should be explained in the main text.

This has been changed to be consistent throughout the entire manuscript and supplementary materials. We now use ‘similar’ and ‘better’ environment hypotheses throughout which we think is much clearer. All tables in Supplementary Figures S1-S6 and tables in the and Table 1 in Main have been edited to reflect these changes.

I was confused in general about this different environment top model – if I read this correctly, the top model is basically the reduced model following model selection criteria?

Yes, this is correct.

But then following Table S2 (and lines 197-200) – wouldn’t that be two models: a reduced model for better environment hypothesis and a reduced model for similar environment hypothesis?

There were two top ranked models and they were for different datasets — non-farmers and farmers, not for the better vs similar environment hypotheses (see Table 1 and Supplementary Materials Tables S2 and S4). In both datasets the similar environment hypothesis is ranked higher. But this was not written very clearly. We have edited lines 197-198 (track changes version) to hopefully help clarify this:

“We include the results for the top ranked models (the similar environment hypothesis) for non-farmers and farmers, respectively, testing each hypothesis alongside those with all variables entered because both approaches have drawbacks.”

Lines 284-285: I think there are some brackets missing here (closing bracket after better env hypothesis and closing bracket after similar env hypothesis).

Thanks. We have added closing parentheses.

Lines 300-303: I understand the rationale here, but these are still interesting results, and I think it would be worth expanding at least a little bit.

We have added the following sentence to line 292-294 track changes version:

“In other words, although it is reasonable to expect that farmers might prefer soil with either more or less peat or a soil type similar to what they were familiar with in Karelia, it is unclear why they would exhibit a preference for a soil composition that was simply different.”

Discussion: I thought the discussion was very thorough and I enjoyed reading it. A few minor comments:

Lines 423-436: It might be just me, but I did not understand the rationale here.

We have added the following sentence to help clarify why evacuees who relocated further away might have been more likely to return home (lines 432-436 track changes version)

“So even though we do not classify geographic distance as a sociocultural variable here, it may serve as a proxy for cultural distance which might help to explain why evacuees who relocated to more distant locations were more likely to return. This is because people arriving from more distant locations are likely to be more culturally different from the host population and may therefore have found it more difficult to assimilate and learn (e.g., acquire new farming techniques) from the host population.”

Lines 474-488: I really appreciated this paragraph, as it echoed some concerns I had reading the paper. Regarding the last line about east-west vs north-south differences, I think it would help if the authors would provide details on how big the differences in environmental conditions (e.g. rainfall or temperature) are in the sample. It might be that these differences are simply not big enough to influence crops or livestock (i.e. the crops might be much more robust compared to the differences recorded in the data, and so it is not that environment does not matter, rather the scale of differences in the sample is too small). I think this is important to mention, as the lack of effect in ecological factors could be attributed to scale.

Yes, good point. We have added table S5 to the Supplementary materials to show the variance (minimum, median and maximum values) of these variables across all Finnish and Karelian municipalities (pasted below) and have referenced it in the main text (lines 307-309):

“See Supplementary Materials Table S5 for descriptive statistics (i.e., range and median values) for all continuous variables used in the study.”

Predictors	Minimum	Median	Maximum
Age in 1942	20	34	72
Geographic distance (km)	14.8	297	809
Linguistic distance (% difference)	0.02	0.31	0.42
Temperature (annual mean centigrade)	5.2	12.0	13.5
Rainfall (mm per year)	405	569	732
Clay (%)	0	11.1	75
Altitude (Median meters)	5	92	428
Lake (%)	0	6.0	66.3
Moraine (%)	0	39.2	76.5
Peat (%)	0	8.6	53.3
Rock (%)	0	11.4	74.1
Farmed (%)	0	9.4	41.9
Livestock per 100 hectares	0	105	11311
Population Density (area/population)	0.11	6.3	1333
Taxes percapita	37.5	127.6	648
Income per capita	4.7	15.8	151

Table S5: Descriptive statistics: Range and median values of all continuous variables used in the study.

References

- Barsbai, T., Lukas, D. & Pondorfer, A. Local convergence of behavior across species. *Science* **371**, 292–295 (2021).
- Bazzi, S., Gaduh, A., Rothenberg, A. D. & Wong, M. Skill Transferability, Migration, and Development: Evidence from Population Resettlement in Indonesia. *Am. Econ. Rev.* **106**, 2658–2698 (2016).
- Honkola, T. *et al.* Evolution within a language: environmental differences contribute to divergence of dialect groups. *BMC Evol. Biol.* **18**, 132 (2018).
- Kettunen, L. Suomen murrekartasto. *Suomen murteet, III. A. murrekartasto.*) Helsinki: Suomalaisen Kirjallisuuden Seura (1940).
- Liebig, T., Puhani, P. A. & Sousa-Poza, A. Taxation and internal migration? Evidence from the Swiss census using community-level variation in income tax rates. *J. Reg. Sci.* **47**, 807–836 (2007).
- Loehr, J., Lynch, R., Mappes, J., Salmi, T. & Pettay, J. Newly Digitized Database Reveals

the Lives and Families of Forced Migrants from Finnish Karelia. *Finnish Yearbook of* (2017).

Lynch, R., Lummaa, V. & Panchanathan, K. Integration involves a trade-off between fertility and status for World War II evacuees. *Nature Human Behaviour* (2019).

Sarvimäki, M., Uusitalo, R. & Jääntti, M. Habit Formation and the Misallocation of Labor: Evidence from Forced Migrations. *Available at SSRN 3361356* (2019)
doi:10.2139/ssrn.3361356.

Sheather, S. *A modern approach to regression with R*. (Springer Science & Business Media, 2009).

Stefanovic, D., Loizides, N. & Parsons, S. Home is where the heart is? Forced migration and voluntary return in turkey's Kurdish regions. *J. Refug. Stud.* **28**, 276–296 (2015).

Tabor, A. S. & Milfont, T. L. Migration change model: Exploring the process of migration on a psychological level. *Int. J. Intercult. Relat.* **35**, 818–832 (2011).

Waris, H. *Siirtoväen sopeutuminen: Tutkimus Suomen karjalaisen siirtoväen sosiaalisesta sopeutumisesta*. vol. 4 (Otava, 1952).

Appendix C

Review RSPB-2021-2298

The authors have done a great job and mitigated most of my concerns related to the novelty of the data and the statistical analysis. Just a few minor clarifications:

You mention in the Response Letter that you add the findings on linguistic similarity of Bazzi et al. 2016 in lines 376to378. These results are cancelled and to not appear in the final draft. I think they fit quite well.

I agree with most of your points regarding Sarvimäki et al., however, I think you should mention their work in lines 140 to 154 of the track changes version. Besides adding them to the discussion, I think it is more transparent to add a few lines right at the beginning when you talk about this data and the natural field experiment setting (who has done similar work and what is the difference). Again, the lines you mention are not precisely where the new paragraph is. Please revise the response letter accordingly.

I like that you refer to your previous work in terms where you define success of migrations. However, for this work you define it as the given opportunity to return. On what is this assumption/definition based? Can you refer to other empirical or theoretical studies? Particularly, you now refer to another work of yours where you give more explanations (which make totally sense to me) but leave out details for this paper. Please clarify the definition better why it is a success to go back.

Thank you for all the effort on the statistical analysis. I am a bit confused now about the data availability on land prior the evacuation. You write now in the manuscript, that *you have coded and entered wheter or not farmers owned any land prior to evacuation*. But in the response letter you say you have only data on land from 1970. Can you just rephrase the first sentence and make it clearer what you did?

The *et al.* are missing in “Barsbai (22), for example, showed that foraging, reproductive and social behaviors of humans, birds...” see line 52 in manscript

Appendix D

Associate Editor Board Member

Comments to Author:

This is a revision of a paper harnessing a 'natural experiment' to test predictors of human migrations. The reviewers and I agree that this is a responsive revision that has greatly improved the paper. However, there are several comments that should still be addressed that arose from the new version of the manuscript. R2 raises two bigger points. The first concerns how ecological parameters relate to farming behavior. While this is likely not be something that can be directly addressed with new analyses, it could be discussed as a limitation of the current study that future work could address. The second comment concerns the demographics of the sample (e.g., were there married female and male farmers included in the sample that were treated as independent in analyses?) and this issue should be explicitly addressed either by confirming that this potential confound is not present in the data or by ensuring that analyses address this potential non-independence of migration decisions concerning married couples. R1 further indicates some relevant citations of relevant prior work that should be integrated, as well as some clarifications about data concerning land availability that should be addressed.

Thank you for your encouraging words and for giving us another opportunity to revise our manuscript. Please see our responses to reviewer #1 for our edited discussion of how ecology relates to farming in Finland and how we deal with the non-independence of farmer couples being sampled twice. Reviewer and editor comments are black and italicized and our response are plain text and green. All references to changes in the manuscript text are quoted, single spaced and italicized.

Reviewer(s)' Comments to Author:

Referee: 1

Comments to the Author(s).

The authors have adequately addressed the major concerns that I had in my initial review. There are a number of minor corrections that should be made for the sake of clarity, and there are two substantive issues that I would also like to raise.

1. The authors should check terminologies closely in the final review to be sure that word choice is standardized throughout. For instance, "environments," I believe is a general reference to both ecological and sociocultural environments. There are a few instances in the paper where it appears that authors use "environment" in specific reference to ecological parameters. Please clearly define the intention for the use of the word "environment" and please make clear when discussing natural environments (i.e., ecologies) or when referencing social environments. Any usage is acceptable, as long as it is clearly defined and used systematically throughout the text.

Thanks for this. We have gone through all instances in which we have used the word 'environment' and have edited accordingly such that when we only mean natural environments we have changed it to 'ecological' and when we mean social environments we write 'social or cultural environments'.

2. *Authors should revisit lines 22-24; 82-83; 748-761 to address confusion in word choice / editing issues.*

We have edited these sections to improve clarity (see new track changes word document lines 24, 83-84 and 537-552).

3. *Lines 238 - 240 More systematic to reference % probability of remaining, to fall in line with previous statements in the text and how data are displayed in the relevant plots.*

We have changed these lines to 'probability of remaining' rather than returning to be consistent.

4. Lines 736 - 747 Relocate some of this explanation to the beginning of the main text, as it is important for the reader to understand the definition of "better" environments, as explained here - particularly the agnosticism of what parameters constitute a "better" ecological condition.

We have edited the text and move this sentence to the last paragraph of the introduction

"The particular ecological conditions that we considered to be 'better' were largely determined by the preferences of farmers as compared to non-farmers (see Methods), as optimal conditions (e.g., more or less rain) largely rely on the type of crop." (Lines 168-171 new track changes word document)

The Methods section (lines 537-552) have been edited to reflect this change and avoid repetition.

5. Line 328, Table 1 H2 is listed as "Different Environments" in the table. Isn't H2 in reference to the Similar Environments Hypothesis? Make sure there is consistency throughout with terminology, as this is confusing.

Yes, this was changed in the latex .tex file and the compiled pdf but we forgot to make the change to the track changes document. We have now done so (Line 309 new track changes word document). The rest of the document has been checked for consistency as well.

6. Line 446 We cannot presume clay soils are worse for farming. Shallow roots and cruciferous vegetables prefer the moisture retention found in clay soils. Again, this muddles an understanding of what a "better" environment is for farmers when authors declare some ecological parameters as favourable compared to others.

Right, this is correct. We had this backwards. Clay soils actually tend to be good for farming in Finland. However, if you move from a non-clay area to an area with a lot of clay your old cultural adaptations will no longer work because clay soils demand their own practices to be

successfully cultivated. Nevertheless, we have removed the reference to clay soil being worse for farming (Lines 416 and 551-552 new track changes word document).

More substantive comments:

1. Although the authors have provided adequate defense of their methodological design and interpretation of results to the extent of the scope of this paper, there is still general weakness in testing the Better Environments Hypothesis. We have no real understanding how the ecological parameters chosen (temperature, altitude, % lake, livestock, farmed land, rainfall, rock, moraine, peat) may bear (or not) on the behaviours of farmers post-relocation, or their adaptive capacity to modify landscapes that circumvent or mitigate any potentially less than optimal farming conditions, thus rendering the ecological differences between origin and destination as insignificant. Furthermore, because ecological data are gathered at the municipality level yet data are analysed at the individual level, it obscures individual variations in plots, where the assumption is that farmers would be choosing to work parcels of land that are optimal within the surrounding areas. In addition to this, the authors have now acknowledged this point; however the point still remains, that the better environments hypothesis presupposes that ecologies have little / no bearing on the outcomes of non-farmers. Although acknowledged, this is somehow still problematic in the analysis.

It is true that we do not have data on how the specific ecological parameters we used impact farmers decisions on what crop to plant or on their ability to modify their fields to mitigate any of the negative effects of non-optimal farming conditions. It is unclear, however, how this type of data could be obtained and analyzed. Although, it might be possible, and would certainly be interesting, to use qualitative self-reports of the evacuees who did change crop type or performed other adaptive modifications in response to what their neighbors were doing, it is beyond the scope of this manuscript. Still, we do not agree that this 'renders the ecological differences between origin and destination location insignificant'. Arguably we have the best possible behavioral data – whether they remained or returned to Karelia (i.e., they voted with their feet). This costly behavior is very likely to be determined in large part by the ability to succeed on the new land, and this is even more likely to be true in the severe climate of Finland where farming operates on the edge of viability. The short growing season leaves little room for planting too early or too late and farmers need to use crops that are suited to the area. Temperature and the predictability of rainfall, in particular, are crucial, and a late frost can wipe out the entire enterprise. Altitude moderates this risk and low peat lands (i.e., former swamps) are the most vulnerable. In this environment where there is a high chance of crop failure, the cultivation of land with a varying amount of peat, clay and moraine all requires specific know how and tools which often require culturally transmitted knowledge from neighbors.

We have added the following to the Discussion section: “Qualitative data might also help add insight into how individuals who remained may have adapted farming practices to local conditions. Still, it is important to note that decisions to return or remain are likely to capture, in large part, a failure to adapt quickly to new conditions which in the harsh climate of Finland (e.g., short growing season and extreme temperatures) can result in the real possibility of total crop failure.” (Lines 446-450 in the new track changes word document)

Yes, it is also true that because the ecological data is collected by municipality the individual variation across plots of lands within these municipalities is obscured. Our analysis therefore assumes that the conditions of individual plots are strongly correlated with the surrounding

areas. While this is very likely to be true of some conditions such as rainfall, altitude, or temperature, it may not always be the case regarding soil conditions. Soil conditions tend to vary with topography, latitude and rainfall, however, so while this is true and ideally, we would have the specific social conditions of each individual plot, we still believe that there is enough consistency across municipalities to make these analyses informative.

We have added the following to the caveats section: *“It should also be noted that because the ecological data is collected at the level of the municipality, the individual variation across plots of lands within these municipalities is obscured. Here we are assuming that the ecological conditions of the individual plots are strongly associated with those of the surrounding areas. While this is almost certainly true of some conditions, including rainfall, temperature, and altitude, this may not always be the case with respect to the specific soil conditions.”* (Lines 450-455 new track changes word document).

In response to the *comment that, ‘the better environments hypothesis presupposes that ecologies have little / no bearing on the outcomes of non-farmers’*, we suggest that although this may be true for some ecological factors, such as temperature, these conditions are unlikely to be as primary as they are in the success of farmers who are far more likely to depend on them for their livelihood. For other ecological conditions, such as soil types, however, it is hard to imagine how non-farmers would be strongly impacted (Lines 254-257 new track changes word document and referenced SI text).

2. The second substantive issue is that there is a demographic problem with the data set. *There is no mention in the text regarding variation in outcomes based on occupation and marital status vis-a-vis gender. Agricultural females are listed as “farmer housewife” in the dataset. Presumably these individuals are married to male farmers who also appear in the same dataset? There may be a handful, but I could not locate any unmarried female farmers. There was a much greater likelihood that female non-farmers (secretaries, nurses, teachers, etc.) were unmarried compared to the female farmers. For the married female farmer population, the decision to return or remain would likely not be independent of her husband's decision to return or remain (as she is almost assuredly housewife to the farmer). Or do we know that all subjects are independent of one another (i.e., no subjects are married)? This would need to be addressed in the analysis.*

This is a good point and something that we were aware of at the beginning of our analysis but then forgot to mention in the paper. We only included the focal individual in our analysis. This was the one member of each household who was interviewed. Although these individuals often provided information on their spouse and children, these other individuals were not included in our analysis. Although most of the detailed information regarding how the data were collected and exclusion criteria were previously moved to the SI, Supplementary Methods section, in an attempt to save space, we have added the following to the Methods section in the main manuscript:

“Our analysis focuses on a subset of individuals who were personally interviewed, one from each household (i.e., the focal interviewee), and on whom complete records were available

which generated 9,870 farmers and 12,204 non-farmers.” (Lines 489-492 new track changes word document).

Referee: 2

*Comments to the Author(s).
see attached comments*

Review RSPB-2021-2298

The authors have done a great job and mitigated most of my concerns related to the novelty of the data and the statistical analysis. Just a few minor clarifications:

You mention in the Response Letter that you add the findings on linguistic similarity of Bazzi et al. 2016 in lines 376 to 378. These results are cancelled and to not appear in the final draft. I think they fit quite well.

We had moved them to the SI to meet Proceedings B space constraints but have now returned them to the main ms (lines 337-339 new track changes word document).

I agree with most of your points regarding Sarvimäki et al., however, I think you should mention their work in lines 140 to 154 of the track changes version. Besides adding them to the discussion, I think it is more transparent to add a few lines right at the beginning when you talk about this data and the natural field experiment setting (who has done similar work and what is the difference). Again, the lines you mention are not precisely where the new paragraph is. Please revise the response letter accordingly.

Yes, we have added the following sentences to the Introduction to acknowledge and contextualize the work of Sarvimäki et al., on this population while both mentioning key differences from our study and referencing the SI for additional context and differences:

“Previous work, using a different dataset on this same population, showed that habits contributed to farmers remaining in agriculture after the war (Sarvimaki, 2019). This analysis, however, relied on the strength of social networks within the evacuee population whereas we are primarily interested in relationships between evacuees and the host population (see SI text: Supplementary Discussion).” (Lines 145-149 new track changes word document)

I like that you refer to your previous work in terms where you define success of migrations. However, for this work you define it as the given opportunity to return. On what is this assumption/definition based? Can you refer to other empirical or theoretical studies? Particularly, you now refer to another work of yours where you give more explanations (which make totally sense to me) but leave out details for this paper. Please clarify the definition better why it is a success to go back.

The assumption we are making is that is a success to remain, not to return. In other words, a successful migration is one where the migrant moves to the new location and is able to remain despite having the ability to return. Gordon Allport (1954) also used this definition to define successful integration into a host community. We are only assuming here that a ‘successful’ migration is one in which the individual remained rather than reverse migration in which they return home as also defined by Lee (1966).

Thank you for all the effort on the statistical analysis. I am a bit confused now about the data availability on land prior the evacuation. You write now in the manuscript, that you have coded and entered whether or not farmers owned any land prior to evacuation. But in the response letter you say you have only data on land from 1970. Can you just rephrase the first sentence and make it clearer what you did?

Yes, okay we can see why this was confusing. We only had data on the amount of land (e.g., number of hectares) the farmers owned in 1970 so we did not use this variable. However, because of the way farmers were labeled in the dataset we did know whether they were landowners (i.e., owned any land at all) prior to the evacuation. We have rephrased this section (in Supplementary Methods: Predictor Variables). It now reads:

“Although we did not have data on how much land individuals owned prior to the evacuation, the amount of land farmers received was commensurate with their holdings in Karelia (Loehr et al., 2017) so failing to add this as a co-variate is unlikely to have a major impact on our results. We did, however, have a simple measure of whether farmers owned any land at all prior to evacuation, but adding this dummy variable to the models did not substantially affect our results and was eliminated in model comparisons (see Model selection below). Therefore, it was not included in any of the final models.”

The et al. are missing in “Barsbai (22), for example, showed that foraging, reproductive and social behaviors of humans, birds...” see line 52 in manuscript.

Yes. We have changed this in the track changes word document (line 73) and in the final latex document (line 52).

References

- Allport, G. W. (1954). *The Nature of Prejudice*, New York: Addison-Wesley.
- Lee, Everett S. 1966. A Theory of Migration. *Demography* 3 (1): 47–57.